# Transcriptional dynamics uncover the role of BNIP3 in mitophagy during muscle remodeling in *Drosophila*

**Hiroki Taoka[1,2†], Tadayoshi Murakawa[3†], Kohei Kawaguchi[1], Michiko Koizumi[1], Tatsuya Kaminishi[4], Yuriko Sakamaki[5], Kaori Tanaka[6], Akihito Harada[6,7], Keiichi Inoue[8], Tomotake Kanki[8], Yasuyuki Ohkawa[6], Naonobu Fujita[1,3]\***

[1]Cell Biology Center, Institute of Integrated Research, Institute of Science Tokyo, Tokyo, Japan; [2]Biological Science Research Laboratories, Kao Corporation, Tokyo, Japan; [3]School of Life Science and Technology, Institute of Science Tokyo, Tokyo, Japan; [4]Department of Genetics, Graduate School of Medicine, Osaka University, Osaka, Japan; [5]Ochanomizu Research Facility (ORF), Bioscience Center, Research Infrastructure Management Center, Institute of Science Tokyo, Tokyo, Japan; [6]Division of Transcriptomics, Medical Institute of Bioregulation, Kyushu University, Fukuoka, Japan; [7]Department of Multi-Omics, Graduate School of Medical Sciences, Kyushu University, Fukuoka, Japan; [8]Department of Cellular Physiology, Graduate School of Medical Sciences, Kyushu University, Fukuoka, Japan

**\*For correspondence:**
nafujita@cbc.iir.isct.ac.jp

[†]These authors contributed equally to this work

**Competing interest:** The authors declare that no competing interests exist.

## eLife Assessment

This paper presents the **important** finding that BNIP3/NIX, a mitophagy receptor, and its binding to ATG18 are required for mitophagy during muscle cell reorganization in Drosophila. Although the involvement of the BNIP3-ATG18/WIPI axis in mitophagy induction has been reported in mammalian cell culture systems, this study provides the first **compelling** evidence for this pathway in vivo in animals. The physiological significance of this BNIP3-dependent mitophagy will require further investigation.

**Abstract** Differentiated muscle cells contain myofibrils and well-organized organelles, enabling powerful contractions. Muscle cell reorganization occurs in response to various physiological stimuli; however, the mechanisms behind this remodeling remain enigmatic due to the lack of a genetically trackable system. Previously, we reported that a subset of larval muscle cells is remodeled into adult abdominal muscle through an autophagy-dependent mechanism in *Drosophila*. To unveil the underlying mechanisms of this remodeling, we performed a comparative time-course RNA-seq analysis of isolated muscle cells with or without autophagy. It revealed both transcriptional dynamics independent of autophagy and highlighted the significance of BNIP3-mediated mitophagy in muscle remodeling. Mechanistically, we found that BNIP3 recruits autophagic machinery to mitochondria through its LC3-interacting motif and minimal essential region, which interact with Atg8a and Atg18a, respectively. Loss of BNIP3 leads to a substantial accumulation of larval mitochondria, ultimately impairing muscle remodeling. In summary, this study demonstrates that BNIP3-dependent mitophagy is critical for orchestrating the dynamic process of muscle remodeling.

## Introduction

Differentiated muscle cells are intricate assemblies of myofibrils and well-organized organelles, such as T-tubules and the sarcoplasmic reticulum. Coordinated sliding of actin and myosin filaments is regulated by the dynamics of calcium ions (Ca²⁺) released from the sarcoplasmic reticulum, while mitochondria provide ATP as the energy source driving this process. Consequently, the structural alignment of organelles and myofibrils within muscle cells is essential for the precise orchestration of contraction (*Kawaguchi and Fujita, 2024*). Muscle cells are long-lived and continuously undergo atrophy and hypertrophy cycles in response to growth factors, mechanical loading, nutrient status, and aging (*Sartori et al., 2021*). Muscle organelles also undergo remodeling as the amount of myofibrils changes, a process mediated by the ubiquitin–proteasome system. However, the mechanisms behind these processes are still poorly understood due to the lack of a genetically tractable system for analyzing muscle remodeling.

Macroautophagy, hereafter referred to as autophagy, is essential for muscle function in both flies and mammals. Autophagy is an intracellular degradation system that delivers cytosolic materials to the lysosome (*Morishita and Mizushima, 2019*; *Nakatogawa, 2020*). A flat membrane sac elongates and encloses cytosolic content to form an autophagosome, a double membrane structure. The completed autophagosome fuses with lysosomes to degrade the enclosed cargo. To date, over 40 *ATG* genes have been identified as key regulators of autophagosome formation and cargo selectivity (*Nakatogawa, 2020*). In mammalian muscles, the loss of autophagy leads to reduced contractile function and abnormalities in myofibrils and organelles, including mitochondria and sarcoplasmic reticulum (*Karsli-Uzunbas et al., 2014*; *Masiero et al., 2009*; *Yoshii et al., 2016*). We have previously reported that a subset of larval body wall muscle cells is remodeled into adult dorsal internal oblique muscles (DIOMs) through an autophagy-mediated mechanism (*Fujita et al., 2017*; *Murakawa et al., 2020*). The remodeled adult DIOMs function during eclosion, persist for approximately 12 hr, and are subsequently eliminated via programmed cell death (*Sánchez-Martín et al., 2019*). To our knowledge, DIOM remodeling is the sole example of developmentally regulated muscle remodeling, making it an ideal model for studying muscle remodeling in general. Autophagy blockade results in severe phenotypes in DIOM (*Fujita et al., 2017*; *Murakawa et al., 2020*), yet the mechanisms by which autophagy loss induces defects remain unclear. Given that autophagy contributes to nutrient supply and transcriptional regulation (*Morishita and Mizushima, 2019*; *Sánchez-Martín et al., 2019*), it is plausible that dysfunction in autophagy could also impact gene expression during DIOM remodeling.

Autophagy can selectively target various substrates, including protein aggregates, invading bacteria into the cytosol, and organelles (*Lamark and Johansen, 2021*; *Vargas et al., 2023*). This selectivity is achieved through soluble and transmembrane cargo receptors, which bridge the cargo and autophagy machinery. Most soluble cargo receptors target ubiquitinated cargoes and harbor oligomerization domains, ubiquitin-binding domains, and LC3/Atg8-interacting regions (LIR), which is a four-residue motif [W/F/Y] -X-X-[L/V/I] (*Johansen and Lamark, 2020*). In contrast, transmembrane cargo receptors are anchored to target organelles and recruit autophagy machinery. Soluble and transmembrane cargo receptors confer selectivity through direct interaction with the ATG machinery. In most cases, in addition to the affinity with Atg8/LC3 via the LIR motif, interaction with core machinery components, such as ULK1, FIP200/Atg11, and Atg9, is required (*Lamark and Johansen, 2021*).

Mitophagy, selective autophagy for mitochondria, is physiologically crucial because mitochondrial malfunction is harmful, resulting in energy insufficiency or excess reactive oxygen species in cells (*Ney, 2015*). In addition to damaged mitochondria, functional mitochondria can be removed by mitophagy to meet distinct cellular demands. Multiple mechanisms exist for mitophagy, including both soluble and transmembrane receptor-mediated mechanisms (*Onishi et al., 2021*). In soluble receptor-mediated mitophagy, the PINK1/Parkin axis labels mitochondria with ubiquitin. The ubiquitinated mitochondria are then recognized by NDP52 and Optineurin, soluble cargo receptors, leading to their sequestration by autophagic membranes (*Narendra and Youle, 2024*). On the other hand, transmembrane cargo receptors are anchored to the outer mitochondrial membrane and recruit autophagy machinery via LIR and other motifs interacting with the ATG machinery (*Onishi et al., 2021*). The concept of receptor-mediated mitophagy was first established in yeast ATG32 studies (*Kanki et al., 2009*; *Okamoto et al., 2009*), and then subsequently, it was found that the basic mechanisms are also conserved in multicellular organisms. The list of mitochondrial outer membrane-embedded cargo receptors is growing: NIX/BNIP3L, BNIP3, FKBP8, FUNDC1, and BCL2L13 (*Onishi et al., 2021*).

Nevertheless, their relative contribution and molecular mechanisms still need to be fully elucidated, especially under physiological conditions. So far, most mitophagy studies have been performed in cultured cells with uncouplers, hypoxia, or genetic perturbations (*Yamashita et al., 2024*).

NIX (also known as BNIP3L) and BNIP3 are homologous receptors for mitophagy in mammals. NIX was initially discovered as a mitophagy receptor during reticulocyte maturation. Seminal studies discovered that NIX is required to clear mitochondria from reticulocytes during differentiation (*Sandoval et al., 2008*; *Schweers et al., 2007*). Following this, its paralog BNIP3 was also shown to regulate mitophagy. NIX and BNIP3 have an LIR motif near their N-terminus and a transmembrane domain (TMD) for localization to the mitochondrial outer membrane near their C-terminus. Thereby, it has been proposed that NIX and BINP3 tether the Atg8/LC3-positive isolation membrane and mitochondria for mitophagy through these domains (*Li et al., 2022*; *Onishi et al., 2021*; *Schwarten et al., 2009*). In addition, Prof. Ney's group in the US identified a minimal essential region (MER), a short motif in NIX required for mitochondria clearance in reticulocytes (*Zhang et al., 2012*). Recently, it has been reported that the MER directly interacts with WIPIs, mammalian Atg18 orthologs, to recruit core autophagy machinery in cultured cells overexpressing BNIP3 (*Adriaenssens et al., 2024*; *Bunker et al., 2023*). Despite these findings, the roles of LIR–Atg8 and MER–Atg18 interactions in BNIP3-mediated mitophagy have not been investigated under physiological conditions.

*Drosophila* BNIP3, the sole ortholog of NIX and BNIP3, contains LIR, MER, and TMD, similar to its mammalian counterparts. In *Drosophila*, it has been reported that BNIP3 plays a role in programmed germline mitophagy required for mitochondrial DNA quality control (*Lieber et al., 2019*; *Palozzi et al., 2022*). However, the mechanism is not well understood. In addition, the contribution of BNIP3 to mitophagy in other physiological contexts, such as cellular differentiation or remodeling programs, has not been explored yet. In this study, we performed a comparative time-course RNA-seq analysis of isolated muscle cells and identified BNIP3-mediated mitophagy as a critical factor in developmentally programmed muscle remodeling. We also explored the underlying mechanism by which BNIP3 tethers mitochondria with the autophagic machinery, as well as the physiological significance of BNIP3-mediated mitophagy in vivo.

## Results

### Transcriptional dynamics of DIOM remodeling during metamorphosis

During DIOM remodeling, autophagy occurs during atrophy, which lasts from 12 hr to 2 d APF (after puparium formation). Then, organelle and myofibril reformation accompanies hypertrophy from 2 to 4 d APF (*Figure 1A*). We previously reported the dynamics of several organelles in DIOM remodeling (*Fujita et al., 2017*; *Murakawa et al., 2020*); however, the overall mechanism is still enigmatic. To gain molecular insights into DIOM remodeling, we performed time-course RNA-seq of isolated DIOMs from four animals at five time points (*Figure 1A*). Dissected animals were fixed with methanol to avoid cell death induction (*Wang et al., 2021*), and six DIOMs were isolated from each animal to prep total RNA (*Figure 1B*). The mRNAs were amplified, and the cDNA libraries were prepared using CEL-Seq2, a sensitive protocol for single-cell RNA-seq (*Hashimshony et al., 2016*). Then, they are subjected to next-generation sequencing. As a result, ~8000 genes were counted in the DIOMs above background (*Supplementary file 1*, ctrl). High similarities among four replicates at each time point show the reproducibility of our protocol (*Figure 1—figure supplement 1A*). We performed DESeq2 principal component analysis (PCA) of all samples to reduce the dimensionality of the data (*Figure 1C*, *Figure 1—figure supplement 1B*). In the PC1–PC2 plane, which explains 43% of expressed genes in DIOMs, the transcriptome changed dynamically from 3IL to 4 d APF (*Figure 1C*). Of note, there was a substantial difference between before (3IL) and after (4 d APF) remodeling in both PC1–2 and PC3–4 (*Figure 1C*), suggesting the functional importance of remodeling larval muscle to adult abdominal muscle during this epoch.

To identify genes with similar temporal expression profiles, we used fuzzy c-means to categorize normalized read counts into 12 clusters (*Figure 1—figure supplement 1C*), each with 379–937 genes (*Figure 1—figure supplement 1D*). The 12 clusters represent unique temporal expression dynamics, with each time point showing a distinct profile of clusters with high expression (*Figure 1D*, *Figure 1—figure supplement 1D*). We also noticed that a time-dependent shift in gene ontology (GO) terms is apparent among the clusters, reflecting the sequential events during DIOM remodeling

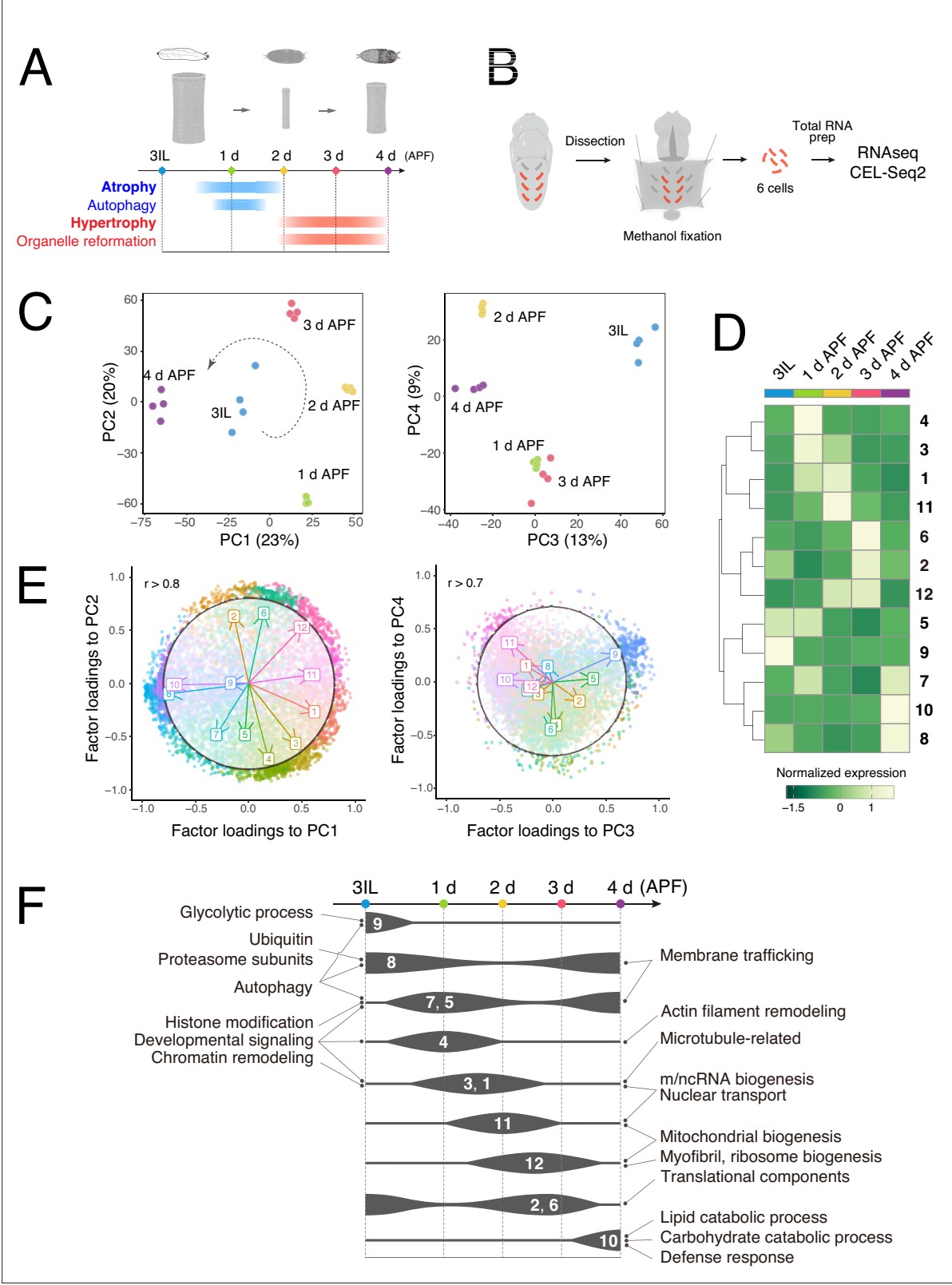

**Figure 1.** Time-course RNA-seq of the dorsal internal oblique muscle (DIOM) remodeling in wild-type *Drosophila*. (**A**) A timeline of DIOM remodeling. Samples were collected at five time points: third instar larva (3IL) and 1–4 days after puparium formation (APF). (**B**) The scheme for sample preparation of DIOMs. Red-colored rectangles indicate single DIOMs. (**C**) DESeq2 principal component analysis (PCA) of all mRNA-seq libraries. The first four principal components are shown. PC1–2, left; PC3–4, right. The dotted arrow in the PC1–2 plot represents the direction of the transcription dynamics.

*Figure 1 continued on next page*

Figure 1 continued

(**D**) Heatmap of fuzzy c-means cluster core expression profiles. All genes were categorized into 12 clusters. (**E**) Factor loadings of each cluster to PC1–2 and PC3–4. The circles show $r = 0.8$ (PC1–2) or $r = 0.7$ (PC3–4). (**F**) Gene ontology (GO) enrichment at each time point during metamorphosis. The width of each line represents the expression level of the clusters indicated.

The online version of this article includes the following figure supplement(s) for figure 1:

**Figure supplement 1.** RNA-seq of dorsal internal oblique muscle (DIOM) remodeling.

(*Figure 1F*). The principal component coefficients (loadings) of individual clusters to PC1–4 are shown in *Figure 1E*. The direction and length of the arrows indicate the contribution of each cluster to principal components (PC1–4). Briefly, upon induction of muscle atrophy (1–2 d APF), the ubiquitin–proteasome system, developmental signaling, autophagy, and histone modification-related genes were upregulated. Then, chromatin remodeling-related genes were elevated around 2 d APF. Myofibril, mitochondrion, and ribosome biogenesis-related genes required for muscle hypertrophy were upregulated around 3 d APF (*Figure 1F*). The data illustrates the sequential events involved in DIOM remodeling, aligning with previously observed morphological changes (*Fujita et al., 2017*; *Murakawa et al., 2020*).

## Transcriptional dynamics occur independently of autophagy during DIOM remodeling

Autophagy deficiency impacts both atrophy and myofibril regeneration during hypertrophy (*Fujita et al., 2017*; *Murakawa et al., 2020*; *Figure 2A*). Given that autophagy is known to influence gene expression through both direct and indirect mechanisms (*Morishita and Mizushima, 2019*; *Sánchez-Martín et al., 2019*), we hypothesized that the loss of autophagy also alters gene expression in DIOMs. To investigate this possibility, we performed a comparative time-course RNA-seq of DIOM remodeling in *Atg18a* RNAi, *FIP200* RNAi, or *Stx17* RNAi DIOMs and compared the results with control data (*Figure 2B*). Atg18a and FIP200 are essential factors for autophagosome formation, and Stx17 is an autophagosomal SNARE protein required for autophagosome–lysosome fusion (*Lőrincz and Juhász, 2020*; *Nakatogawa, 2020*). Thus, their knockdown blocks autophagy at early or late stages, respectively (*Figure 2B*, bottom). Similar to the experiment in *Figure 1*, we performed RNA-seq at five time points from 3IL to 4 d APF (*Figure 2B*) and confirmed high similarities among the four replicates in all 20 conditions (*Figure 2—figure supplement 1A* and *Supplementary file 1*). We confirmed that RNAi efficiently knocked down targeted genes (*Figure 2—figure supplement 1B–D*). In contrast to our prediction, the knockdown of *Atg18a*, *FIP200*, or *Stx17* only had a slight impact on transcriptomic dynamics in DIOM remodeling (*Figure 2C*), with only minor changes detected (*Figure 2—figure supplement 2G*).

Next, to examine whether loss of autophagy affects protein synthesis in DIOMs, we used the quantification of DIOM volume changes as a surrogate measurement and compared controls with *FIP200* RNAi from 3IL to 4 d APF (*Figure 2D, E*). As expected, muscle atrophy observed until 2 d APF was affected by *FIP200* RNAi (*Figure 2A, D*). Notably, DIOMs in the *FIP200* RNAi condition grew from 2 to 3 d APF at levels comparable to the control (*Figure 2E*), indicating that protein synthesis occurs independently of autophagy. Consistent with this observation, transcription of both ribosomal RNAs (rRNAs) and proteins, established indicators of the target of rapamycin complex 1 (TORC1) activity (*Saxton and Sabatini, 2017*), did not drop but instead was slightly elevated in conditions lacking autophagy (*Figure 2—figure supplement 2*). The synthesis of rRNA was upregulated in *ATG* RNAi between 2 and 3 d APF (*Figure 2—figure supplement 2A*). It was confirmed that the number of nuclei remained unchanged by the loss of autophagy (*Figure 2—figure supplement 2B, C*). The size of the nucleolus, which positively correlates with TORC1 activity (*Grewal et al., 2007*), was increased by *FIP200* and *Stx17* RNAi (*Figure 2—figure supplement 2D–F*). Moreover, most ribosomal protein large subunit (RpLs) and ribosomal protein small subunit (RpSs) mRNA levels were higher in *ATG* RNAi conditions (*Figure 2—figure supplement 2G*). These results suggest that TORC1 activity and protein synthesis capacity are comparable between control and autophagy-deficient DIOMs. Altogether, we conclude that gene expression occurs independently of autophagy in DIOMs.

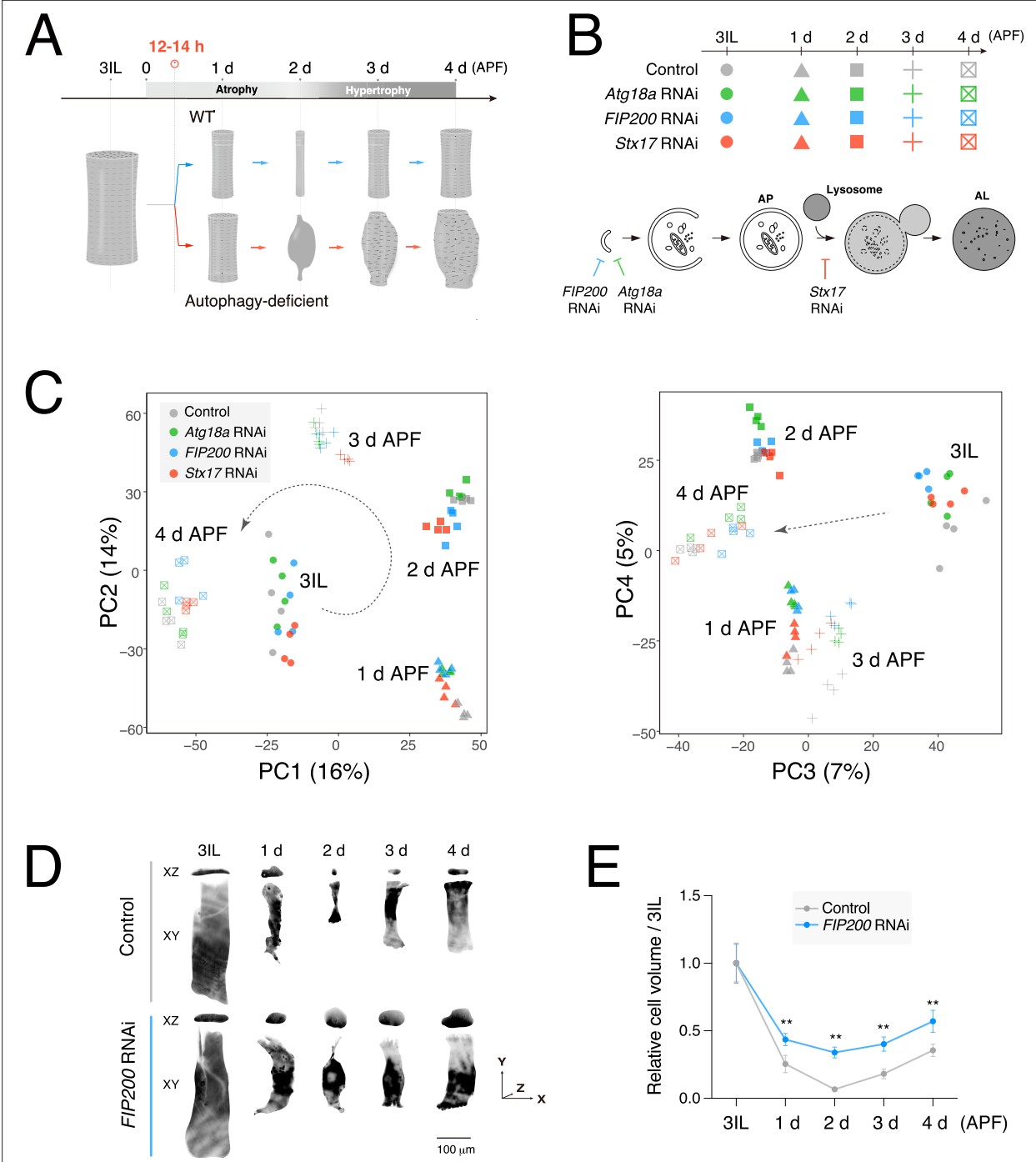

**Figure 2.** Transcriptional dynamics during dorsal internal oblique muscle (DIOM) remodeling is independent of autophagy. (**A**) A schematic of DIOM remodeling in wild-type and autophagy-deficient conditions. Autophagy-dependent muscle atrophy starts at 12–14 hr APF. (**B**) Genotypes and time points of the comparative RNA-seq analysis (top) and a diagram of autophagosome formation (bottom). *FIP200* or *Atg18a* RNAi blocks autophagosome formation. *Stx17* RNAi blocks the autophagosome–lysosome fusion. (**C**) DESeq2 principal component analysis (PCA) of all mRNA-seq libraries. PC1–2, left; PC3–4, right. A total of 20 samples were analyzed. (**D, E**) DIOM volume changes in control or *FIP200* RNAi from 3IL to 4 d APF. DIOMs were labeled with GFP. Projected images of XY and XZ planes are shown (**D**). (**E**) Relative DIOM cell volume for each genotype normalized to 3IL (set to 1). *N* = 5 (Mann–Whitney test). **p < 0.001.

The online version of this article includes the following figure supplement(s) for figure 2:

**Figure supplement 1.** Expression levels of Atg18a, FIP200, and Stx17 during dorsal internal oblique muscle (DIOM) remodeling.

**Figure supplement 2.** Effect of autophagy deficiency on transcription of ribosomal RNA (rRNA) and proteins.

## Loss of BNIP3 leads to an accumulation of mitochondria in DIOMs

The above results suggest that the autophagic degradation of cytosolic components in itself is critical for DIOM remodeling. We have previously reported that *FIP200* RNAi or *Stx17* RNAi induce the accumulation of mitochondria or mitophagosomes, respectively (*Figure 3A, B*; *Fujita et al., 2017*). In addition, mitophagosomes were often observed in wild-type DIOMs at 1 d APF, when autophagy is strongly induced (*Figure 3C*; *Fujita et al., 2017*). These results suggest that mitochondria are major autophagy cargo, and impaired mitophagy contributes to the severe loss of autophagy phenotype.

We tried to identify relevant mitophagy receptors directly via our RNA-seq approach. The time-course RNA-seq data (*Figures 1 and 2*) indicated that, among the known mitophagy regulators, only BNIP3 was robustly expressed in 1 d APF DIOMs. In contrast, *Zonda*, *CG12511*, *Pink1*, *Park*, *Key*, *Ref(2)P*, and *IKKε*—the *Drosophila* orthologs of FKBP8, FUNDC1, PINK1, Parkin, Optineurin, p62, and TBK1, respectively—showed little or undetectable expression at this stage (*Figure 3D*). BNIP3 is the sole fly ortholog of mammalian NIX and BNIP3. *BNIP3* mRNA levels were relatively high at 3IL and 4 d APF when mitophagy was minimally induced. We noticed that *FBXL4* is expressed in 3IL and 4 d APF but not in 1 d APF DIOM (*Figure 3E*). BNIP3 is known to undergo ubiquitination and degradation through a mechanism mediated by FBXL4, a ubiquitin E3 ligase (*Niemi and Friedman, 2024*). Previous data suggest that transcriptional and post-translational mechanisms regulate BNIP3 protein levels. Consistent with expression levels, *BNIP3* RNAi led to mitochondrial accumulation in 4 d APF DIOMs, whereas RNAi targeting other known mitophagy regulators did not (*Figure 3—figure supplement 1*). We generated *BNIP3* knockout (KO) flies lacking all exons using two guide RNAs (gRNAs) to characterize BNIP3 further (*Figure 3F*). *BNIP3* KO was compatible with viability and did not markedly impair mobility (*Figure 3—figure supplement 2A–C*). Similar to RNAi, mitochondria accumulated significantly in *BNIP3* KO DIOMs at 4 d APF (*Figure 3G*).

At the ultrastructural level, *BNIP3* KO induced the accumulation of mitochondria and loss of myofibrils (*Figure 4A–C*), consistent with our confocal data (*Figure 3G*). To test the role of BNIP3 in mitophagosome formation more directly, we performed transmission electron microscopy (TEM) in the absence of Stx17 to block the autophagosome–lysosome fusion. As expected, mitophagosomes (pink-colored) were frequently observed in the *Stx17* RNAi condition (*Figure 4D, D', F*). In stark contrast, mitophagosomes were rare, and unengulfed mitochondria (green-colored) accumulated in the combined condition of *Stx17* RNAi and *BNIP3* KO (*Figure 4E, E', F*). We also confirmed there was no significant difference in the number of autophagosomes (blue-colored) between the two conditions (*Figure 4G*), indicating that BNIP3 is dispensable for autophagosome formation. These data show that BNIP3 is critical for mitophagosome formation in DIOMs.

## The LIR and MER motifs in BNIP3 are required for the clearance of mitochondria in DIOMs

*Drosophila* BNIP3 harbors LIR, MER, and TMD domains, similar to mammalian NIX and BNIP3 (*Figure 5A*). Although the function of the MER domain has been unknown since it was identified (*Zhang et al., 2012*), recent studies have shown that the MER of mammalian NIX and BNIP3 directly interacts with WIPIs, the mammalian orthologs of Atg18 (*Adriaenssens et al., 2024*; *Bunker et al., 2023*). AlphaFold 3 (*Abramson et al., 2024*) predicted that this interaction is evolutionarily conserved in *Drosophila* in a manner analogous to their mammalian counterparts (*Figure 5B*; *Adriaenssens et al., 2024*; *Bunker et al., 2023*). In this prediction, residues 31–57 of *Drosophila* BNIP3 (slate blue and yellow) interact with the surface formed by blades 2 (green) and 3 (pink) of the β-propeller structure of *Drosophila* Atg18a. Notably, residues 42–53 (yellow), which correspond to the MER designated for NIX (*Bunker et al., 2023*; *Zhang et al., 2012*), fold into an α-helix, docking into the groove formed by blades 2 and 3 and forming substantial hydrophobic interactions as well as defined polar contacts. We confirmed that HA-tagged *Drosophila* Atg18a co-immunoprecipitated with GFP-tagged full-length *Drosophila* BNIP3, and that this interaction was attenuated by the deletion of the MER (residues 42–53) (*Figure 5C*).

Next, to verify the importance of the LIR and MER motifs in BNIP3, we expressed a series of BNIP3 mutants in *BNIP3* KO flies (*Figure 5A*). ΔLIR (W16A/L19A) lacks critical consensus residues in LIR due to double mutations (*Johansen and Lamark, 2020*). The MER mutant (MER[mut], L49A) substituted residue L49 with an A, which is predicted to be deeply buried at the interface with Atg18a (*Figure 5C*). The corresponding L75A mutant in human NIX has been shown to lose its affinity for

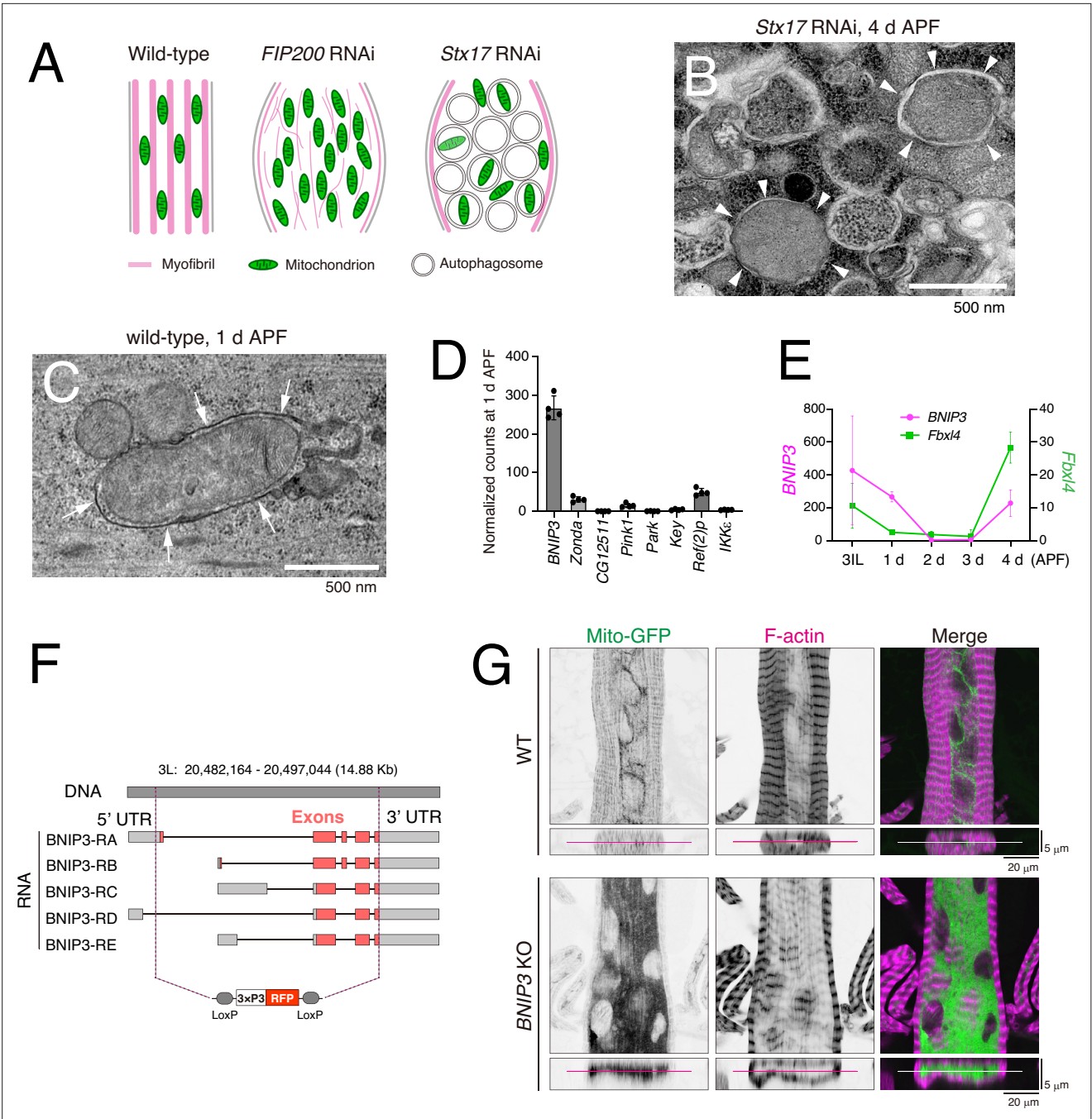

**Figure 3.** Loss of BNIP3 results in accumulation of mitochondria in dorsal internal oblique muscles (DIOMs). (**A**) A schematic of 4 d APF DIOMs in wild-type, *FIP200* RNAi, or *Stx17* RNAi. (**B**) Transmission electron microscopy (TEM) image of *Stx17* RNAi DIOM at 4 d APF. White arrowheads indicate autophagosome structures. (**C**) TEM image of wild-type DIOM at 1 d APF. White arrows indicate mitophagosome membrane structure. (**D**) The expression level of mitophagy regulators in DIOMs at 1 d APF. *N* = 4. Normalized counts in RNA-seq are shown. (**E**) The expression level of *BNIP3* and *Fbxl4* in DIOMs. *N* = 4. (**F**) A diagram illustrating the *BNIP3* knockout strategy, where all exons were deleted using two guide RNAs (gRNAs) and replaced with a 3xP3-RFP marker. (**G**) Loss of BNIP3 phenotype on mitochondria and myofibrils in DIOM at 4 d APF. XY (top) and XZ (bottom) planes were shown.

The online version of this article includes the following figure supplement(s) for figure 3:

**Figure supplement 1.** Loss of BNIP3 on mitochondria and myofibrils in dorsal internal oblique muscles (DIOMs).

**Figure supplement 2.** Loss of BNIP3 on adult fly lifespan and mobility.

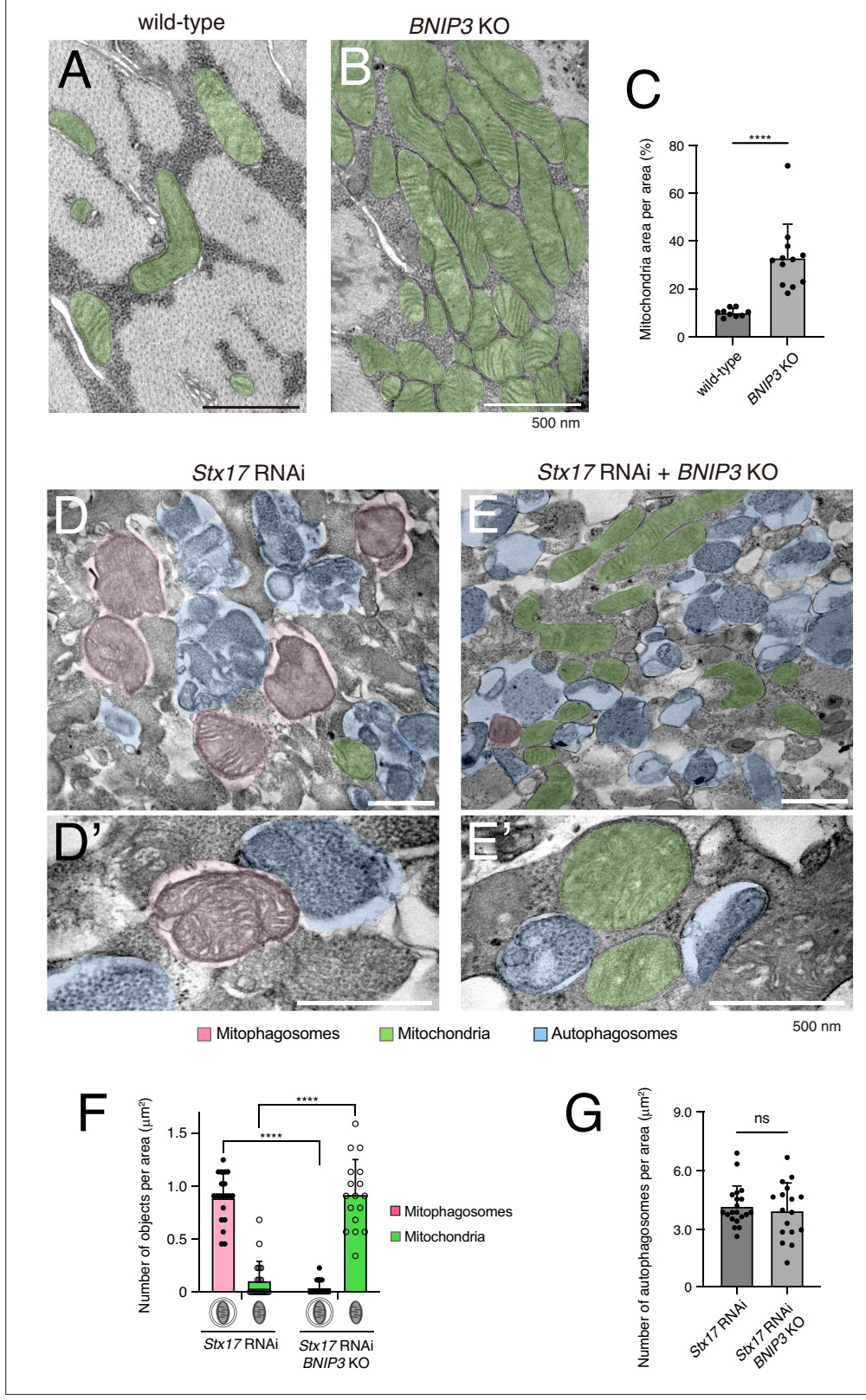

**Figure 4.** BNIP3 is required for mitophagosome formation. Transmission electron microscopy (TEM) images of dorsal internal oblique muscle (DIOM) transverse sections at 4 d APF in wild-type (**A**) or *BNIP3* KO (**B**). Mitochondria are shown in green. (**C**) The percentages of mitochondrial area per unit area in the indicated genotypes. Wild-type, *N* = 9; *BNIP3* KO, *N* = 12 (Mann–Whitney test). ****p < 0.00001. (**D and E**) TEM images

*Figure 4 continued on next page*

*Figure 4 continued*

of DIOM transverse sections at 4 d APF in *Stx17* RNAi (**D, D'**) or a combination of *Stx17* RNAi and *BNIP3* KO (**E, E'**). Mitophagosomes, pink; Mitochondria, green; autophagosomes, blue. (**F**) The number of mitophagosomes and mitochondria per unit area in the indicated genotypes. (**G**) The number of total autophagosomes, including mitophagosomes, per unit area in the indicated genotypes. *Stx17* RNAi, N = 20; *Stx17* RNAi and *BNIP3* KO, N = 17 (Mann–Whitney test) (**F, G**).

WIPI2/Atg18 (*Bunker et al., 2023*). The ΔMER mutant lacks an entire short α-helix (G42 to Q53) of the MER motif (*Figure 5C*, shown yellow-orange). The ΔLIR + ΔMER is a combined construct of ΔLIR and ΔMER. All BNIP3 constructs retain an intact TMD at the C-terminus, ensuring their localization to the outer mitochondrial membrane (*Figure 5—figure supplement 1A*).

As shown in *Figure 5D*, the expression of the full-length (Full) construct almost completely suppressed the accumulation of mitochondria in BNIP3 KO at 4 d APF, confirming the rescue of BNIP3 function using our construct-based approach. To our surprise, the BNIP3 ΔLIR construct rescued the *BNIP3* KO phenotype comparable to Full, indicating LIR on its own may be redundant for mitophagy in DIOMs. In contrast, mitochondria accumulated in MER^mut and ΔMER reconstituted DIOMs. Furthermore, the combination of ΔLIR and ΔMER resulted in mitochondria accumulation nearly identical to that observed in BNIP3 KO flies (*Figure 5D, D'*). Consistent with these findings, *Atg101*-dependent autophagic degradation of GFP-BNIP3 was strongly suppressed by the ΔLIR + ΔMER mutations (*Figure 5E*). Atg101 is an essential subunit of the Atg1/ULK1 kinase complex (*Guo et al., 2019*; *Hegedűs et al., 2014*; *Noda and Mizushima, 2016*). From these results, we conclude that both MER and LIR motifs contribute to BNIP3-mediated mitophagy, with some redundancy in their contributions to selectivity.

## BNIP3-mediated mitophagy eliminates larval muscle mitochondria during muscle remodeling

The data above suggest that a BNIP3-mediated mechanism degrades larval muscle-derived mitochondria; however, in previous figures, we only observed the terminal phenotype of *BNIP3* KO at 4 d APF. To observe the changes in the number of mitochondria and cell shape during DIOM remodeling, we first performed a time-course experiment in control and *BNIP3* KO conditions (*Figure 6A*, *Figure 6—figure supplement 1A, B*). There was no significant difference in the number of mitochondria at 3IL; however, mitochondria started accumulating at 1 d APF (*Figure 6A, A'*), suggesting that BNIP3-mediated mitophagy degrades larval muscle mitochondria. Further, we tested the Mito-QC probe (*Lee et al., 2018*), a tandem GFP-mCherry fusion protein targeted to the OMM, to compare mitophagy flux at 1d APF with and without BNIP3. When the probe is delivered to lysosomes via autophagy, the GFP is quenched due to the acidic environment of the lysosome. Robust mitophagy activity was detected in control DIOMs at 1 d APF. In sharp contrast, *BNIP3* KO blocked mitophagy flux at a comparable level with *FIP200* RNAi (*Figure 6B, B'*).

To show the degradation of larval muscle-derived mitochondria more directly, we labeled larval muscle mitochondria specifically by using the temperature-sensitive mutant of GAL80 (GAL80^ts), a repressor of GAL4 (*McGuire et al., 2004*). At the restrictive temperature (29°C), the GAL80^ts mutant is inactive, allowing GAL4 to drive the expression of the UAS-fused construct, UAS-Mito-GFP. Conversely, at the permissive temperature (18°C), the GAL80^ts is active, suppressing GAL4 activity. Using this system, Mito-GFP was expressed in the larval muscle and subsequently suppressed throughout the entire pupal stage (*Figure 6C*). Anti-ATP5A antibodies were used to visualize total mitochondria. The amount of Mito-GFP-positive mitochondria was comparable in control and *BNIP3* KO at 3IL muscle. However, at 4 d APF, the Mito-GFP signal was nearly absent in the control, while a strong Mito-GFP signal was observed in the *BNIP3* KO (*Figure 6D, D'*), indicating that BNIP3-mediated mitophagy degrades larval muscle mitochondria during muscle remodeling.

## Discussion

The transcriptional dynamics associated with DIOM remodeling are largely independent of autophagy (*Figure 2*). Instead, our RNA-seq data suggest that it is regulated primarily by ecdysone signaling, with minimal influence from autophagy inhibition. However, a subset of genes exhibited altered

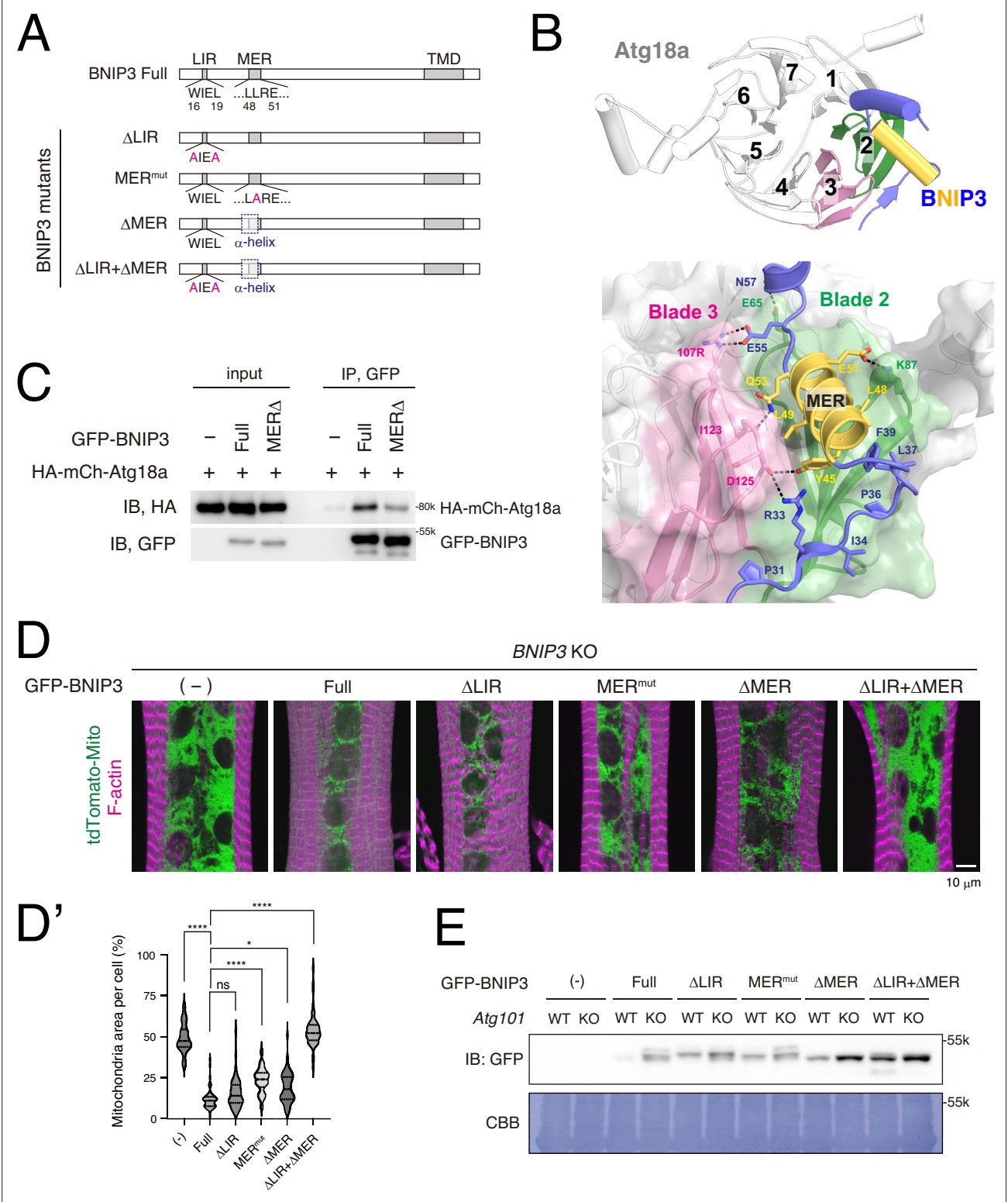

**Figure 5.** The LIR and MER motifs are required for BNIP3-mediated mitochondrial clearance. (**A**) Schematics of *Drosophila* BNIP3 and its mutants. (**B**) The structure of the BNIP3-Atg18a complex predicted by AlphaFold 3. Top, overview: Atg18a is depicted in white, featuring a β-propeller structure consisting of seven blades, with blades 2 and 3 highlighted in green and pink, respectively. For BNIP3, only residues 29–74 are shown in blue for clarity, with the α-helix spanning residues 42–53 highlighted in yellow. Bottom, close-up view: Amino acids positioned to form intramolecular contacts through their side chains are labeled and represented as sticks, with potential hydrogen bonds shown as dashed lines. (**C**) GFP pulldown experiment between

*Figure 5 continued on next page*

*Figure 5 continued*

GFP-BNIP3_full or MERΔ construct and HA-mCh-Atg18a in HEK293 cells. (**D, D'**) BNIP3 rescue experiment in dorsal internal oblique muscles (DIOMs) at 4 d APF. The indicated GFP-tagged BNIP3 constructs and tdTomato-Mito were co-expressed in *BNIP3* KO flies using the GAL4/UAS system (**D**). Single-channel images corresponding to the merged panels are presented in *Figure 5—figure supplement 1B*. The percentages of mitochondrial area in DIOMs for the indicated genotypes. Empty (-), *N* = 46; Full, *N* = 55; ΔLIR, *N* = 52; MER$^{mut}$, *N* = 48; ΔMER, *N* = 54; ΔLIR + ΔMER, *N* = 51 (Kruskal–Wallis test) *p < 0.05 and ****p < 0.00001 (**D'**). (**E**) The amount of GFP-BNIP3 in *WT* and *Atg101* KO muscles. The GFP-BNIP3 constructs were expressed by Mef2-GAL4 in *WT* or *Atg101* KO flies. Larval fillets were lysed and subjected to western blotting for GFP. After immunoblotting, the membrane was stained with CBB to assess total protein loading.

The online version of this article includes the following source data and figure supplement(s) for figure 5:

**Source data 1.** PDF files containing original western blots for *Figure 5C, E*.

**Source data 2.** Original files for western blot analysis displayed in *Figure 5C, E*.

**Figure supplement 1.** GFP-BNIP3 constructs localize to the outer mitochondrial membrane.

**Figure supplement 2.** Effect of GFP-BNIP3 overexpression on mitophagy flux in 3IL BWMs.

expression levels in autophagy-deficient conditions (*Figure 2—figure supplement 2G*). Of note, their expression levels were consistently upregulated or downregulated throughout all time points. Since no apparent phenotype was observed in *ATG* knockdown 3IL body wall muscles (*Fujita et al., 2017*), the observed expression changes are unlikely to underlie the loss-of-autophagy phenotype in DIOM remodeling. Furthermore, *FIP200* RNAi had minimal impact on the DIOM growth rate (*Figure 2D, E*), which reflects translational capacity. From these data, we conclude that the contribution of autophagy to gene expression during DIOM remodeling is minimal.

In the RNA-seq experiments (*Figure 2*), *ATG* genes were exclusively depleted in muscle cells; thus, autophagy in non-muscle organs remained intact. It is plausible that essential nutrients for DIOM hypertrophy, such as amino acids, were supplied by other organs, including the fat body (*Murakawa et al., 2022*). Alternatively, the necessary amino acids might have been derived from myofibril breakdown via the ubiquitin–proteasome system (*Quy et al., 2013*).

Transcriptome changes before and after DIOM remodeling reflect adaptations to the shift from the anaerobic environment of larvae to the aerobic environment of adults, alongside alterations in nutritional pathways. Notable changes in energy production systems occurred between the 3IL stage (pre-remodeling) and 4 d APF (post-remodeling) (*Figure 1*). Glycolysis-associated genes exhibited high expression levels at the 3IL stage, followed by a significant decline during the pupal and adult stages. For instance, the expression of *Lactate dehydrogenase* (*Ldh*), a key enzyme in switching between anaerobic and aerobic respiration (*Liang et al., 2023*; *Semenza, 2014*), was particularly noteworthy. Ldh catalyzes the conversion of pyruvate to lactate while regenerating NAD$^+$ from NADH under anaerobic conditions. Ldh levels were markedly high in larval muscles, suggesting reliance on anaerobic glycolysis. Conversely, its reduced expression in remodeled adult muscles reflects a metabolic shift toward aerobic pathways.

Genes related to serine-centered amino acid and lipid catabolism, both of which rely on oxygen to produce ATP, were significantly upregulated in adult muscles. For example, the expression of *Alanine-glyoxylate aminotransferase* (*Agxt*), an enzyme that converts alanine to pyruvate, was approximately 100 times higher in adult muscles compared to larvae. Similarly, *brummer* (*bmm*), a triacylglycerol lipase, exhibited a fivefold increase, suggesting enhanced reliance on fatty acids metabolized via β-oxidation and the TCA cycle. These findings highlight the shift in energy sources from glycolysis in larval muscles to amino acids and lipids in adult muscles, metabolized aerobically in mitochondria.

A recent study suggests that DIOMs are a primary source of pupal ecdysone (*Zhang et al., 2024b*). In line with this data, our time-course mRNA-seq analysis revealed high expression of *Phm*, an enzyme essential for ecdysteroid biosynthesis, at 1 d APF (*Supplementary file 1*). However, other Halloween genes required for ecdysteroid biosynthesis, such as *Nvd*, *Spo*, *Spok*, *Dib*, *Sad*, and *Shd* (*Kamiyama and Niwa, 2022*), were not, or only minimally, expressed in DIOMs during metamorphosis (*Supplementary file 1*). Consequently, whether DIOMs directly secrete ecdysone remains unclear. Instead, DIOMs may contribute to a specific step in the ecdysteroid biosynthesis pathway, potentially supplying intermediate 5β-ketotriol to other tissues for further conversion.

Functional mitochondria are degraded during developmental muscle remodeling; therefore, it is reasonable that a membrane-anchored mitophagy receptor, directly interacting with the autophagic

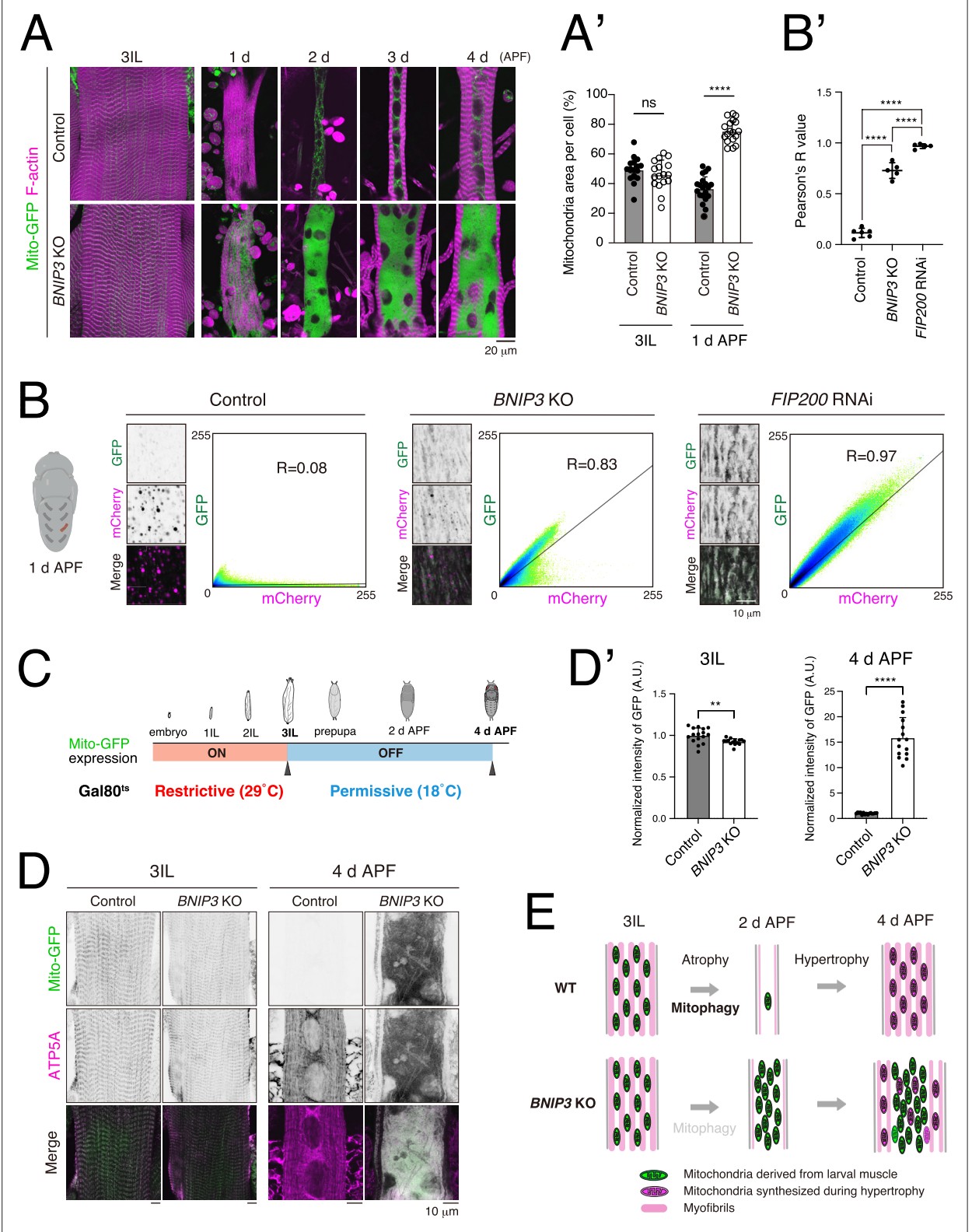

**Figure 6.** BNIP3-mediated mitophagy eliminates larval muscle mitochondria during muscle remodeling. (**A, A'**) Time-course microscopy of Mito-GFP and F-actin in control or *BNIP3* KO during dorsal internal oblique muscle (DIOM) remodeling (**A**). Single-channel images corresponding to the merged panels are presented in *Figure 6—figure supplement 1A*. Mitochondrial area relative to total cell area. For 3IL, control, *N* = 18; *BNIP3* KO, *N* = 17. For 1 d APF, control, *N* = 20; *BNIP3* KO, *N* = 18 (Mann–Whitney test) ****p < 0.00001 (**A'**). (**B, B'**) Mitophagy assay using Mito-QC in DIOMs at 1 d APF. Pixel

*Figure 6 continued on next page*

*Figure 6 continued*

intensity correlation profiles and Pearson's correlation coefficients (*R* values) are shown. Control, *N* = 6; *BNIP3* KO, *N* = 5. *FIP*200 RNAi, *N* = 5 (Sidak's test) ****p < 0.00001. (**C**, **D**, **D′**) Scheme of the use of GAL80 temperature-sensitive mutants (GAL80$^{ts}$). The animals were raised at 29°C (Restrictive) to induce Mito-GFP expression until mid-3IL, then shifted to 18°C (Permissive) to block expression (**C**). Mito-GFP and ATP5A immunostaining (total mitochondria) signals in muscles at 3IL or 4 d APF (**D**). Mito-GFP intensities in muscles at each time point normalized to control (set to 1). For 3IL, control, *N* = 16; *BNIP3* KO, *N* = 15. For 1 d APF, control, *N* = 15; *BNIP3* KO, *N* = 15 (Mann–Whitney test) **p < 0.001 and ****p < 0.00001 (**D′**). (**E**) A model of muscle remodeling with or without BNIP3. Loss of BNIP3 leads to mitochondrial accumulation, which disrupts muscle remodeling.

The online version of this article includes the following figure supplement(s) for figure 6:

**Figure supplement 1.** The impact of BNIP3 knockout on the shape of the dorsal internal oblique muscle (DIOM) during metamorphosis.

machinery, is selected over a ubiquitination-mediated mechanism involving PINK1 and Parkin. The PINK1/Parkin axis is activated by mitochondrial membrane potential depolarization (*Narendra and Youle, 2024*), which is not observed in remodeling muscles. The protein level of BNIP3 appears to be regulated at transcriptional and post-translational levels (*Figure 3E*). It is known that BNIP3 is degraded via a mechanism mediated by Fbxl4, a ubiquitin E3 ligase (*Niemi and Friedman, 2024*). The balance between synthesis and degradation likely results in the highest BNIP3 protein levels at 1 d APF, a time when mitophagy is robustly induced. However, BNIP3 protein levels have not been directly confirmed, as isolating a sufficient number of DIOMs for western blotting is challenging. Generating a GFP knock-in line for BNIP3 would allow for an imaging-based assay to measure its protein levels during DIOM remodeling.

Overexpression of BNIP3 in larval muscle cells did not significantly induce mitophagy (*Figure 5— figure supplement 2*), suggesting that BNIP3 expression alone is insufficient to drive mitophagy. BNIP3-mediated mitophagy likely requires not only BNIP3 expression but also autophagy induction. We propose that autophagy induction in DIOMs at the early pupal stage is regulated by ecdysone signaling, similar to other instances of developmental autophagy in *Drosophila* (*Murakawa et al., 2022*). Understanding the mechanisms by which developmental signals trigger autophagy remains a critical question for future research.

A key function of autophagy in DIOM remodeling is the degradation of mitochondria from larval muscles. In the absence of BNIP3, mitochondria derived from the larval muscle accumulate and cluster in the cell center, physically obstructing myofibril formation during hypertrophy and restricting myofibrils to the cell periphery (*Figure 6E*). Interestingly, unlike *ATG* RNAi, *BNIP3* knockout muscles exhibited relatively normal but thinner myofibrils (*Figure 3G*). In contrast, *ATG* RNAi not only caused mitochondrial accumulation but also severely disrupted myofibril organization, as indicated by misaligned sarcomeres (*Fujita et al., 2017*; *Murakawa et al., 2020*). These findings suggest that autophagy contributes to DIOM remodeling beyond mitochondrial degradation. Schnorrer's group in France has demonstrated that mechanical tension is essential for sarcomere assembly and maturation (*Mao et al., 2022*; *Zhang et al., 2024a*). Given that *ATG* RNAi disrupts DIOM attachment (*Murakawa et al., 2020*; *Ribeiro et al., 2011*), it is possible that reduced tension—caused by an unknown mechanism—leads to disorganization of myofibrils in DIOMs when autophagy is impaired.

Contrary to the previous model, the LIR motif of BNIP3 was found to be dispensable for mitophagy in DIOMs (*Figure 5D, E*). Our findings align with reports stating that the clearance of mitochondria in reticulocytes is independent of the LIR motif (*Novak et al., 2010*; *Zhang et al., 2012*). Together, this suggests that the dependency on the LIR motif may differ between in vitro and in vivo contexts. In studies of BNIP3 in cultured cells, such as HeLa cells, BNIP3 is often overexpressed, and mitophagy is induced through chemical treatments (*Bunker et al., 2023*; *Yamashita et al., 2024*). These experimental conditions may account for the observed differences in LIR dependency. Understanding the factors that drive these contextual variations in BNIP3-mediated mitophagy will require further investigation.

The MER motif in BNIP3 is required for mitophagy in *Drosophila*. Similar to its mammalian counterpart, the MER motif of *Drosophila* BNIP3 is predicted to interact with the groove between blades 2 and 3 of the β-propeller in Atg18a (*Figure 5B*). Notably, this groove of Atg18 overlaps with the binding site for Atg16 (*Strong et al., 2021*). We propose that the BNIP3 MER–Atg18a interaction functions at the early stages of mitophagosome formation, which is later replaced by the Atg16–Atg18a interaction during the elongation phase. In this sequential model, the BNIP3 LIR–Atg8a interaction would dominate at the elongation steps. Alternatively, the MER–Atg18a and LIR–Atg8a interactions might

serve redundant roles in determining selectivity, as both Atg8 and Atg18 localize to the elongating autophagic membranes. This redundancy could explain why the LIR motif is dispensable for BNIP3-mediated mitophagy (*Figure 5D, E*).

Martens's group (Austria) recently reported that WIPIs, the mammalian orthologs of Atg18, bind to Atg13, a component of the FIP200/ULK1 complex (*Adriaenssens et al., 2024*). They propose that the NIX/BNIP3–WIPI–Atg13 axis induces selective autophagy for mitochondria, analogous to known selective autophagy receptors associated with FIP200 (*Lamark and Johansen, 2021*). It would be interesting to test whether an artificial interaction motif with the Atg1 complex could compensate for the loss of function of ΔMER in DIOMs.

Our study provides new insights into the mechanisms underlying autophagy-associated muscle remodeling. Transcriptional dynamics uncovered a sequence of events involving muscle atrophy and hypertrophy, offering a valuable dataset for understanding the process of muscle remodeling. In addition, we provided insights into the mechanism and significance of BNIP3-mediated mitophagy in vivo. These findings are particularly important given the evolutionary conservation of muscle remodeling mechanisms between *Drosophila* and mammals (*Piccirillo et al., 2014*). Future studies should explore the broader implications of our findings across various cellular, developmental, and organismal contexts.

## Materials and methods

### Reagents and antibodies

The following antibodies were used: Mouse monoclonal anti-ATP5A (1:300, ab14748, Abcam, Cambridge, UK), Rat monoclonal anti-HA (1:1000; 11867423001, Roche, Basel, Switzerland), Rabbit polyclonal anti-GFP (1:2000; 598, MBL, Nagoya, Japan), Rabbit polyclonal anti-Fibrillarin (1:300; ab5821, Abcam, Cambridge, UK), anti-rabbit IgG Alexa Fluor 594 conjugate (1:400; A11012, Thermo Fisher Scientific, Waltham, USA), anti-mouse IgG Alexa Fluor 488 conjugate (1:400; A11001, Thermo Fisher Scientific, Waltham, USA), anti-mouse IgG Alexa Fluor 594 conjugate (1:400; A11005, Thermo Fisher Scientific, Waltham, USA), HRP-conjugated AffiniPure Goat Anti-Rabbit IgG (1:10,000; 111-035-144, Jackson ImmunoResearch, West Grobe, USA), HRP-conjugated AffiniPure Donkey Anti-Rat IgG (1:5000; 712-005-153, Jackson ImmunoResearch, West Grobe, USA), Bisbenzimide H33342 Fluorochrome Trihydrochloride DMSO Solution (1:10,000; 04915-81, Nacalai Tesque, Kyoto, Japan), Alexa Fluor 633 Phalloidin (1:200; A22284, Thermo Fisher Scientific, Waltham, USA), and GFP-Trap Agarose (gta, Chromotek, Planegg-Martinsried, Germany).

### *Drosophila* strains

Flies were reared under standard conditions at 25°C unless otherwise stated. The $w^{1118}$ strain was used as a control for knockout strains. For muscle-targeted gene expression, Mef2-GAL4 was used. UAS-*LacZ* was used as a control for UAS transgenes. All genetic combinations were generated by standard crosses. Detailed genotypes are described in *Supplementary file 3*. Genotypes of flies used in this study include the following: (1) $y^1$ $w^*$; P{$w^{+mC}$ = GAL4-Mef2.R}3 (Bloomington *Drosophila* Stock Center, BL 27390; DMef2-GAL4), (2) $w^{1118}$; P{$w^{+mC}$ = UAS-lacZ.B}Bg4-1-2 (BL 1776; UAS-LacZ), (3) $w^*$; P{$w^{+mC}$ = UAS-Rheb.Pa}2 (BL 9688), (4) $w^{1118}$; P{$w^{+mC}$ = UAS-mito-HA-GFP.AP}2/CyO (BL 8442; Mito-GFP), (5) $w^{1118}$; P{$y^{+t7.7}$ $w^{+mC}$ = UAS-mito-QC}attP16 (BL 91640, Mito-QC), (6) $w^{1118}$; P{$w^{+mC}$ = UAS-Dcr-2.D}10 (BL 24651, UAS-Dcr2), (7) $y^*$ $w^*$; P{$w^{+mC}$ = UAS-tdTomato.mito}2 (Kyoto DGGR 117016, tdTomato-Mito), (8) $w$; UAS-IR-Atg18a$^{KK100064}$ (VDRC 105366; Atg18a RNAi), (9) $w$; UAS-IR-Stx17$^{KK100034}$ (VDRC 108825; Stx17 RNAi), (10) $w$; UAS-IR-BNIP3$^{KK111246}$ (VDRC 107493; BNIP3 RNAi), (11) $w$; UAS-IR-Pink1$^{KK101205}$ (VDRC 109614; Pink1 RNAi), (12) $w$; UAS-IR-Park$^{KK107919}$ (VDRC 104363; Park RNAi), (13) $w$; UAS-IR-CG12511$^{KK109618}$ (VDRC 101785; CG12511 RNAi), (14) $w$; UAS-IR-Zonda$^{KK107775}$ (VDRC 110620; Zonda RNAi), (15) $y^1$ $sc^*$ $v^1$ $sev^{21}$; P{$y^{+t7.7}$ $v^{+t1.8}$=TRiP.HMS01611}attP2/TM3, $Sb^1$ (TRiP, BL 36918; FIP200 RNAi), (16) $y^1$ $v^1$; P{$y^{+t7.7}$ $v^{+t1.8}$=TRiP.JF01937}attP2 (TRiP, BL 25896; Stx17 RNAi), (17) $y^1$ $sc^*$ $v^1$ $sev^{21}$; P{$y^{+t7.7}$ $v^{+t1.8}$=TRiP.GL00156}attP2 (TRiP, BL 35578; mTor RNAi), (18) $w$; UAS-tub-GAL80$^{ts}$ (from E. Kuranaga), (19) $w$; UAS-mCD8:GFP, (20) $w$; DMef2-GAL4, UAS-Dcr2 (*Fujita et al., 2017*), (21) $w$; UAS-Dcr2; DMef2-GAL4, sqh-YFP:Mito$^{BL7194}$/TM6C Sb Tb (*Murakawa et al., 2020*). New genotypes generated during this study include the following: (22) $w^*$;; BNIP3 KO CRISPR{3xP3-RFP}/TM6B, $Tb^1$, (23) $w^{1118}$, Atg101 KO CRISPR{3xP3-RFP}/FM7a, (24) $w^{1118}$; UASt-GFP-BNIP3_full$^{attP40}$, (25) $w^{1118}$;

*UASt-GFP-BNIP3_ΔLIR (W16A/L19A)*$^{attP40}$, (26) *w$^{1118}$; UASt-GFP-BNIP3_MER mutant (L49A)$^{attP40}$*, (27) *w$^{1118}$; UASt-GFP-BNIP3_ΔMER (Δ42–53)* $^{attP40}$, and (28) *w$^{1118}$; UASt-GFP-BNIP3_ ΔLIR+ΔMER* $^{attP40}$.

## DNA engineering

Plasmid vectors were constructed using standard molecular biology techniques. The DNA polymerase KOD One (TOYOBO, Osaka, Japan) and PrimeSTAR GXL Premix (Takara Bio, Shiga, Japan) were used for PCR amplification. The DNA fragments were assembled using the Gibson Assembly. Construction of the expression vectors of GFP-tagged BNIP3_full under UAS control (pUASt-attB-GFP-BNIP3) was as follows: *Drosophila* BNIP3 CDS was amplified from template RE48077 (DGRC Stock 9148; https://dgrc.bio.indiana.edu//stock/9148; RRID:DGRC_9148) using the following primer sets: 5′-ATGT CTACGACACCAAAATCGAG-3′ and 5′-TCAGTCAATGACCACACGG-3′. This fragment of BNIP3 CDS was ligated into the linearized pUAST-attB-GFP vector, which is amplified with the following primer sets: 5′-CGTGTGGTCATTGACTGAGCGGCCGCGGCTCGAGGGTACC-3′ and 5′-TTGGTGTCGTAG ACATGGTGAAGGGGGCGGCCGCGGAG-3′. To generate plasmids with mutations or deletions in the BNIP3 coding sequence (pUAST-attB-BNIP3_ΔLIR, pUAST-attB-BNIP3_MER$^{mut (L49A)}$, and pUAST-attB-BNIP3_ΔMER), the plasmid pUAST-attB-GFP-BNIP3_full was used as a template. For the construction of pUAST-attB-BNIP3_ΔLIR, these primer sets: 5′-CTGCGATCGAAGCGAGCACAACAGCTGCGATG -3′ and 5′-CTCGCTTCGATCGCAGATTCGCCCAGCAAATC-3′ were used to introduce the W16A/L19A mutations. For the construction of pUAST-attB-BNIP3 MER$^{mut (L49A)}$, these primer sets: 5′-AGACTTGC GCGCGAGGCCCAGCGCGAG-3′ and 5′-CTCGCGCGCAAGTCTCAGGTACTCCTC-3′ were used to introduce the L49A mutation. For the construction of pUAST-attB-BNIP3_ΔMER, these primer sets: 5′-CAACAATCGCGAGTCGAACCAGTCG-3′ and 5′-GACTCGCGATTGTTGAATGGCAACGG-3′ were used to delete G42 to Q53. To generate a construct that harbors both ΔLIR and ΔMER (pUAST-attB-BNIP3_ΔLIR+ΔMER), the plasmid pUAST-attB-GFP-BNIP3_ΔMER was used as a template. The same primer sets as pUAST-attB-BNIP3_ ΔLIR construction were used for amplification. Each resultant linear DNA fragment was ligated into a circular plasmid. Generation of the mammalian expression constructs—pMRX-IRES-puro-GFP-BNIP3_full, pMRX-IRES-puro-GFP-BNIP3_ΔMER, and pMRX-IRES-puro-3xHA-mCh-Atg18a—was performed by amplifying the corresponding inserts from pUAST-attB-GFP-BNIP3_full, pUAST-attB-GFP-BNIP3_ΔMER, or LD38705 (DGRC Stock 2722; https://dgrc.bio.indiana.edu//stock/2722; RRID:DGRC_2722), respectively. The amplified fragments were subsequently ligated into the linearized pMRX-IRES-puro vector. All the resultant vectors were validated by DNA sequencing.

## Generation of mutant and transgenic flies

To generate *Drosophila* mutants with a knockout of *BNIP3* (CG5059) or *Atg101* (CG7053), the CRISPR/Cas9 system was used, following a modified method based on *Kondo and Ueda, 2013*. gRNA sets for *BNIP3* KO and *Atg101* KO were designed as follows: for *BNIP3* KO, upstream gRNA sequences, 5′-CGTCAACACCAAAGATAACT[TGG]-3′ and downstream gRNA sequences 5′-CTTTCAGTCAAT GACCACAC[GGG]-3′; for *Atg101 KO*, upstream gRNA sequences 5′-GTCCACCTGACGACCCTCCA [TGG]-3′ and downstream gRNA sequences 5′-CTCGCAATGTGACGGGCTGT[CGG]-3′. Each gRNA was individually cloned into a plasmid under the control of the U6 promoter. A donor DNA cassette containing a 3x P3-RFP selection marker, loxP sites, and two homology arms was cloned into the pUC57-Kan plasmid for DNA repair. The gRNAs targeting *BNIP3* or *Atg101*, and the hs-Cas9 coding DNA plasmids, along with the donor plasmid, were microinjected into embryos of the *w$^{1118}$* strain. F1 files were screened by a selection marker of 3x P3-RFP and validated by genomic PCR and sequencing (WellGenetics Inc). To generate a series of UASt-GFP-BNIP3 transgenic flies, the vectors described above were injected into embryos for phiC31-mediated insertion (WellGenetics Inc).

## RNA-seq analysis of DIOMs

For collecting DIOMs or their precursors, larvae or pupae were dissected and fixed with methanol. Six DIOMs were isolated from each animal using forceps and a pipette under a stereomicroscope (SZX16, EVIDENT, Tokyo, Japan). The collected muscle cells were lysed with BLA buffer (ReliaPrep Tissue RNA Miniprep System, Promega) and stored at –80°C. Following sample preparation, the frozen samples were thawed, and total RNA was extracted and eluted in RNase-free water according to the manufacturer's protocol (ReliaPrep Tissue RNA Miniprep System, Promega). Subsequently,

cDNA libraries were prepared using CEL-Seq2, a single-cell RNA-seq protocol (*Hashimshony et al., 2016*). Briefly, reverse transcription was performed using poly(T) primers with a T7 promoter to generate single-stranded DNA from mRNA, which was then converted into double-stranded DNA through second-strand synthesis. Amplified RNA was produced via in vitro transcription using the MEGAscript T7 kit (Thermo Fisher Scientific, Waltham, MA) and subsequently reverse transcribed to construct the cDNA library. Library sequencing was performed using the NovaSeq6000. Cell barcode and UMI in Read1 were extracted by using UMI-tools with the following command 'umi_tools extract --bc-pattern=CCCCCCCCCCCCCCCCNNNNNNNNNN --stdin read1 --stdout read1_out --read2-in read2 --read2-out=read2_out --whitelist=whitelist'. To extract barcodes, umi_tools whitelist was used. Barcodes in the whitelist were validated by comparing them to barcodes used in the reverse transcription. Reads were mapped to the BDGP6 reference using HISAT2. Read counts for each gene were obtained by featureCounts. The expression levels of genes in each sample were determined from the UMI counts after a normalization process using the DESeq2 package in R. The R programming language was used for each type of analysis, utilizing the pheatmap package for cluster analysis, the prcomp package for PCA, and the Mfuzz package for fuzzy c-means clustering analysis (*Kumar and Futschik, 2007*).

## Quantification of rRNA in DIOMs
Total RNA was extracted as described in the 'RNA-seq analysis of DIOMs' section. The quality and composition of the extracted RNA were assessed using the Agilent 2100 Bioanalyzer system with the RNA 6000 Pico Kit (Agilent, Santa Clara, USA). rRNA content in the DIOMs was quantified by measuring the peak areas corresponding to 18S and 28S rRNA within the total RNA profile.

## Immunofluorescence staining
Muscle preparations were performed as previously described (*Ribeiro et al., 2011*). 3IL or pupae were pinned on a Sylgard-covered Petri dish in dissection buffer (5 mM HEPES, 128 mM NaCl, 2 mM KCl, 4 mM MgCl$_2$, 36 mM sucrose, pH 7.2). The animals were pinned flat and fixed (4% PFA, 50 mM EGTA, PBS) at room temp for 20 min. Then, the samples were unpinned and blocked (0.3% bovine serum albumin, 0.6% Triton X-100, PBS) at room temp for 5 min, incubated with primary antibody overnight at room temperature, washed with PBS, and incubated with Alexa Fluor 488 or 594-conjugated secondary antibody (Thermo Fisher Scientific, Waltham, MA) for 2 hr at room temperature. The immunostained samples were washed and mounted in FluorSave reagent (Merck Millipore, Darmstadt, Germany).

## Confocal fluorescence microscopy
To image live pupal DIOMs, staged pupae were removed from their pupal cases and mounted between a slide glass and a cover glass following the protocol described by *Zitserman and Roegiers, 2011*. Imaging was performed through the dorsal cuticle. Both live and fixed samples were observed using a confocal microscope (FV3000, EVIDENT, Tokyo, Japan) equipped with either a ×60 silicone/1.30 NA UPlanSApo or a ×4/0.16 objective lens (EVIDENT, Tokyo, Japan). FLUOVIEW (EVIDENT, Tokyo, Japan) was used for image acquisition, and the exported images were processed and analyzed with ImageJ (NIH, Bethesda, USA).

## Transmission electron microscopy
Staged pupae (1 or 4 d APF) were removed from pupal cases, pinned on a Sylgard-covered Petri dish, and dissected directly in fixative (2% paraformaldehyde, 2.5% glutaraldehyde, 150 mM, 5 mM calcium chloride, sodium cacodylate, pH 7.4). They were then fixed for 2 hr at room temperature followed by overnight at 4°C. The dissected fillets were washed with 0.1 M phosphate buffer pH 7.4, post-fixed in 1% OsO$_4$ buffered with 0.1 M phosphate buffer for 2 hr, dehydrated in a graded series of ethanol, and embedded flat in Epon 812 (TAAB, Aldermaston, UK). Ultrathin sections (70 nm thick) were collected on copper grids covered with Formvar (Nisshin EM, Tokyo, Japan), double-stained with uranyl acetate and lead citrate, and then observed using a JEM-1400Flash transmission electron microscope (JEOL, Tokyo, Japan).

## Immunoblots and immunoprecipitation

For immunoblots, five larval fillets were collected for each sample. The samples were lysed directly in 1.5× sample loading buffer [100 mM Tris-HCl (pH 6.8), 3% SDS, 15% glycerol, and 0.0015% Bromophenol Blue]. Equal amounts of proteins per sample were subjected to SDS–PAGE and transferred to Immobilon-PSQ PVDF transfer membrane (Merck Millipore, Darmstadt, Germany) with Trans-Blot Turbo Transfer System (Bio-Rad, Hercules, CA, USA). The membranes were blocked with TBS-T buffer [10 mM Tris-HCl (pH 8.0), 150 mM NaCl, 0.1% Tween 20, PBS] containing 5% skim milk for 1 hr, and were then incubated with primary antibodies in Can Get Signal Solution 1 (TOYOBO, Osaka, Japan) overnight at 4°C. Membranes were washed three times with TBST, incubated with HRP-conjugated secondary antibodies in the blocking buffer for 2 hr at room temperature, and washed five times with TBST. Immunoreactive bands were detected using Clarity Western ECL Substrate (Bio-Rad, Hercules, CA, USA) and the ChemiDoc MP Imaging System (Bio-Rad, Hercules, CA, USA). The images were processed using ImageJ.

For immunoprecipitation, HEK293 cells were transfected with plasmids using jetPRIME transfection reagent (Polyplus-transfecton, Illkirch, France) according to the manufacturer's instructions. The following plasmids were used: pMRX-IRES-puro (-), pMRX-IRES-puro-GFP-BNIP3_full (Full), pMRX-IRES-puro-GFP-BNIP3_ΔMER (MERΔ), and pMRX-IRES-puro-3xHA-mCh-Atg18a. After 24 hr of transfection, cells were washed with PBS and lysed in lysis buffer [20 mM HEPES (pH 7.4), 1 mM EDTA, 150 mM NaCl, 0.1% Triton X-100, 10% glycerol, 1 mM PMSF, and protease inhibitor cocktail (Nacalai Tesque, Kyoto, Japan)]. Cell lysates were centrifuged at 17,000 × $g$ for 1 min at 4°C, and the supernatants were incubated with GFP-Trap agarose beads (Chromotek, Planegg-Martinsried, Germany) for 1.5 hr at 4°C with gentle rotation. After incubation, beads were washed with Wash Buffer [20 mM HEPES (pH 7.4), 150 mM NaCl, 0.1% Triton X-100, 10% glycerol] to remove non-specific proteins. Bound proteins were eluted by boiling in 2× sample buffer for 5 min, and subsequent steps were performed as described above for immunoblots.

## Structure prediction and visualization

The amino acid sequences of *Drosophila* BNIP3 (Q9VPD6) and Atg18a (Q9VSF0) were retrieved from the UniProt Knowledgebase (https://www.uniprot.org/uniprotkb/) and submitted to the AlphaFold Server (https://alphafoldserver.com/) to predict the structure of their complex. The PyMOL Molecular Graphics System (Version 2.5.4; Schrödinger, LLC) was then used to analyze the resulting five predictions and visualize the top-ranked structure.

## Adult lifespan assay

The Adult lifespan assay was performed according to a previously reported protocol (*Linford et al., 2013*). Briefly, age-synchronized adult flies were collected, sorted by sex, and housed in groups of 30 per vial. Flies were maintained at 25°C under a 12 hr light/12 hr dark cycle and transferred to fresh food vials every 2–3 days without anesthesia. Mortality was recorded at each transfer, and statistical significance was determined using the log-rank test.

## Fly climbing assay

The climbing assay was conducted according to the protocol previously published (*Gevedon et al., 2019*). Adult flies were gently transferred without anesthesia into a vertical climbing chamber composed of two connected vials, marked at 5 cm from the bottom. After a brief recovery period, flies were tapped to the bottom of the chamber, and the number of flies that climbed above the 5 cm mark within 10 s was recorded. The climbing success rate was calculated as the percentage of flies that reached or surpassed the mark. Each group was tested in five trials, with a rest period of several minutes between trials. The median success rate across trials was used as the representative value for each group.

## Eclosion assay

Late-stage pupae were collected and maintained at 25°C. The eclosion process was recorded, and the time from initial rupture of the pupal case to the complete emergence of the adult fly was measured. For each group, 7–10 individuals were analyzed, and the average elapsed time was calculated.

## Image analyses

For cell volume measurement (*Figure 2E*), the DIOMs expressing GFP were imaged at 1 µm intervals. The muscle cell volume was calculated by multiplying the area of each XY slice by the interval length and summing the results across all slices. To measure nucleolus and nucleus volume (*Figure 2—figure supplement 2E, F*), DIOMs stained for DNA (H33342) and Fibrillarin were imaged at 0.5 µm intervals. DAPI- and anti-Fibrillarin-stained areas were extracted by binarization using ImageJ. This process was repeated for the z-stacks, and the nucleolus and nucleus volume were determined by multiplying the area of each slice by the interval length between slices. For mitochondrial area measurement in TEM images (*Figure 4C*), each mitochondrion and the corresponding DIOM region were manually segmented, and the total mitochondrial area and DIOM area per image were calculated for each image using ImageJ. The total mitochondrial area was then normalized to the DIOM area. For each genotype, at least nine images from multiple animals were analyzed. Quantification of the number of autophagosomes, mitophagosomes, or mitochondria in TEM images was conducted as follows (*Figure 4D, E*): Double-membrane-bound structures containing undigested cytoplasmic contents (autophagosomes), autophagosomes enclosing mitochondria (mitophagosomes), and unengulfed mitochondria were manually counted. At least 17 images derived from multiple animals were analyzed for each genotype. Mitochondrial area in DIOMs is quantified in *Figure 5D' and 6A', D'*. Cellular regions containing F-actin were manually segmented, and tdTomato (*Figure 5D'*) or GFP (*Figure 6A', D'*) channel intensity within these segmented regions was measured. For *Figures 5D' and 6A'*, image preprocessing steps, including filtering, background subtraction, and intensity binarization, were applied to the tdTomato (*Figure 5D'*) or GFP (*Figure 6A'*) channel using ImageJ to reduce noise. The area of each detected object was measured, and the total mitochondrial area was defined as the sum of these object areas. Mitochondrial area values were then normalized to the corresponding cellular region area. For *Figure 6D'*, GFP intensity quantification was performed as follows: because GFP signals in the 3IL BWM images were weaker than those at 4 d APF, the GFP channel intensity was uniformly enhanced across all images at a constant scale. All images were preprocessed using filtering and background subtraction. The mean GFP intensity within each segmented area was then measured and normalized, with the mean GFP intensity of the control for each developmental stage set to 1.

For *Figure 5D'*, at least 46 DIOMs from 5 animals were analyzed per genotype. For *Figure 6A', D'*, at least five DIOMs per fly were analyzed, with three biological replicates per genotype.

For the Mito-QC assay, a representative DIOM was imaged for each genotype and analyzed by ImageJ (*Figure 6B*). Image preprocessing included filtering and background subtraction for both GFP and mCherry channels. Dot plots were then generated, and Pearson's correlation coefficients for GFP and mCherry intensities were calculated using the *Coloc2* plugin in ImageJ (*Figure 6B, B'*).

## Statistics

Each experiment was performed with at least three different cohorts of unique flies analyzed. One exception was for TEM analyses, performed on two parallel replicates with multiple animals each. All experiments were performed in parallel with controls. Raw data files corresponding to all quantified results are provided in *Supplementary file 2*. Error bars show the standard deviation for bar charts. When more than two genotypes were used in an experiment, Sidak's test, Kruskal–Wallis test, or Dunnett's T3 multiple comparisons test was used (GraphPad, Prism9 version 9.5.1). Mann–Whitney tests were used to compare two means. $p < 0.05$ was regarded as statistically significant (*$p < 0.05$, **$p < 0.001$, ***$p < 0.0001$, and ****$p < 0.00001$).

## Acknowledgements

We are grateful to JH Lee (Univ of Michigan), T Yoshimori (Osaka Univ), E Kuranaga (Kyoto Univ), BDSC, VDRC, DGRC, and NIG-fly for reagents. We thank Y Shimada and R Niwa (Univ of Tsukuba) for the helpful discussion. We thank the Biomaterials Analysis Division of the Institute of Science Tokyo for the DNA sequencing. We are grateful to M Landekic (McGill Univ) for English editing. This work was supported in part by Grant-in-Aid for Transformative Research Areas (B) (grant number 21H05147, NF), Japan Science and Technology Agency (JST) PRESTO (grant number JPMJPR18H8, NF), AMED PRIME (grant number 24gm6410016h0001, NF), and MEXT Promotion of Development of a Joint Usage / Research System Project: Pan-Omics DDRIC, MRCI for High Depth Omics, CURE: JPMXP1323015486 for MIB, RIIT, and AMRC in Kyushu University.

## Additional information

### Funding

| Funder | Grant reference number | Author |
|---|---|---|
| Japan Society for the Promotion of Science | 21H05147 | Naonobu Fujita |
| Japan Science and Technology Agency | 10.52926/JPMJPR18H8 | Naonobu Fujita |
| Japan Agency for Medical Research and Development | 24gm6410016h0001 | Naonobu Fujita |
| Ministry of Education, Culture, Sports, Science, and Technology | JPMXP1323015486 | Yasuyuki Ohkawa |

The funders had no role in study design, data collection, and interpretation, or the decision to submit the work for publication.

### Author contributions

Hiroki Taoka, Data curation, Formal analysis, Investigation, Visualization, Writing – review and editing; Tadayoshi Murakawa, Data curation, Software, Formal analysis, Validation, Investigation, Visualization, Methodology, Writing – review and editing; Kohei Kawaguchi, Michiko Koizumi, Yuriko Sakamaki, Kaori Tanaka, Keiichi Inoue, Investigation, Writing – review and editing; Tatsuya Kaminishi, Investigation, Visualization, Writing – review and editing; Akihito Harada, Resources, Investigation, Writing – review and editing; Tomotake Kanki, Conceptualization, Writing – review and editing; Yasuyuki Ohkawa, Resources, Writing – review and editing; Naonobu Fujita, Conceptualization, Resources, Data curation, Formal analysis, Supervision, Funding acquisition, Investigation, Visualization, Writing – original draft, Project administration, Writing – review and editing

### Author ORCIDs

Hiroki Taoka ![ORCID] http://orcid.org/0009-0008-1198-3790
Tadayoshi Murakawa ![ORCID] http://orcid.org/0009-0003-1376-8772
Tatsuya Kaminishi ![ORCID] https://orcid.org/0000-0001-5056-1074
Yasuyuki Ohkawa ![ORCID] https://orcid.org/0000-0001-6440-9954
Naonobu Fujita ![ORCID] https://orcid.org/0000-0003-1914-8438

Reviewer #1 (Public review): https://doi.org/10.7554/eLife.105834.3.sa1
Reviewer #2 (Public review): https://doi.org/10.7554/eLife.105834.3.sa2
Reviewer #3 (Public review): https://doi.org/10.7554/eLife.105834.3.sa3
Author response https://doi.org/10.7554/eLife.105834.3.sa4

## Additional files

### Supplementary files

Supplementary file 1. Normalized counts from time-course RNA-seq of dorsal internal oblique muscle (DIOM) remodeling during metamorphosis.

Supplementary file 2. Raw data files corresponding to all quantified results presented in the manuscript.

Supplementary file 3. Detailed *Drosophila* genotypes, developmental stages, and experimental temperatures are provided.

MDAR checklist

### Data availability

RNA-seq data have been deposited in GEO under accession code GSE293359.

The following dataset was generated:

| Author(s) | Year | Dataset title | Dataset URL | Database and Identifier |
|---|---|---|---|---|
| Murakawa T, Fujita N, Ohkawa Y, Tanaka K, Harada A, Taoka H | 2025 | Comparative Time-Course RNA-Seq Analysis of Abdominal Muscle cell Remodeling in Drosophila with or without Autophagy | https://www.ncbi.nlm.nih.gov/geo/query/acc.cgi?acc=GSE293359 | NCBI Gene Expression Omnibus, GSE293359 |

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

# Appendix 1

### Appendix 1—key resources table

| Reagent type (species) or resource | Designation | Source or reference | Identifiers | Additional information |
|---|---|---|---|---|
| Gene (*Drosophila melanogaster*) | BNIP3 | GenBank | FLYB:FBgn0037007 | |
| Gene (*Drosophila melanogaster*) | Atg18a | GenBank | FLYB:FBgn0035850 | |
| Gene (*Drosophila melanogaster*) | Atg17 (FIP200) | GenBank | FLYB:FBgn0037363 | |
| Gene (*Drosophila melanogaster*) | Stx17 | GenBank | FLYB:FBgn0035540 | |
| Gene (*Drosophila melanogaster*) | Atg101 | GenBank | FLYB:FBgn0030960 | |
| Strain, strain background (*Escherichia coli*) | DH5α | Nippon Gene | Cat# 314-06234 | Competent cells |
| Antibody | anti-ATP5A (Mouse monoclonal) | Abcam | Cat# ab14748, RRID:AB_301447 | IF (1:300) |
| Antibody | anti-HA (Rat monoclonal) | Roche | Cat# 11867423001, RRID:AB_390918 | WB (1:1000) |
| Antibody | anti-GFP (Rabbit polyclonal) | MBL | Cat#598, RRID:AB_591816 | WB (1:2000) |
| Antibody | anti- Fibrillarin (Rabbit polyclonal) | Abcam | Cat# ab5821, RRID:AB_2105785 | IF (1:300) |
| Antibody | anti-rabbit IgG Alexa Fluor 594 conjugate (Goat polyclonal) | Thermo Fisher Scientific | Cat# A11012, RRID:AB_2534079 | IF (1:400) |
| Antibody | anti-mouse IgG Alexa Fluor 488 conjugate (Goat polyclonal) | Thermo Fisher Scientific | Cat# A11001, RRID:AB_2534069 | IF (1:400) |
| Antibody | anti-mouse IgG Alexa Fluor 594 conjugate (Goat polyclonal) | Thermo Fisher Scientific | Cat# A11005, RRID:AB_2534073 | IF (1:400) |
| Antibody | HRP-conjugated AffiniPure Goat Anti-Rabbit IgG (Goat polyclonal) | Jackson ImmunoResearch | Cat# 111-035-144, RRID:AB_2307391 | WB (1:10,000) |
| Antibody | AffiniPure Donkey Anti-Rat IgG (Donkey polyclonal) | Jackson ImmunoResearch | Cat# 712-005-153, RRID:AB_2340631 | WB (1:5000) |
| Recombinant DNA reagent | RE48077 (plasmid) | DGRC | DGRC Stock Number:9148 RRID:DGRC_9148 | BNIP3 cDNA |
| Recombinant DNA reagent | pUASt-attB-GFP (plasmid) | This paper | | Backbone for UAS constructs with GFP. |
| Recombinant DNA reagent | pUASt-attB-GFP-BNIP3_full (plasmid) | This paper | | Plasmid construct of GFP-tagged full-length BNIP3 under UAS control. |
| Recombinant DNA reagent | pUAST-attB-BNIP3_ΔLIR (plasmid) | This paper | | GFP-BNIP3 construct with mutations in LIR motif (W16A/L19A). |
| Recombinant DNA reagent | pUAST-attB-BNIP3_MERmut (L49A) (plasmid) | This paper | | GFP-BNIP3 construct with mutation in MER motif (L49A). |

*Appendix 1 Continued on next page*

*Appendix 1 Continued*

| Reagent type (species) or resource | Designation | Source or reference | Identifiers | Additional information |
|---|---|---|---|---|
| Recombinant DNA reagent | pUAST-attB-BNIP3_ΔMER (plasmid) | This paper | | GFP-BNIP3 construct with deletion in MER motif (G42 to Q53). |
| Recombinant DNA reagent | pUAST-attB-BNIP3_ ΔLIR+ΔMER (plasmid) | This paper | | GFP-BNIP3 construct with mutations in LIR and deletion in MER motifs. |
| Recombinant DNA reagent | pMRX-IRES-puro-GFP-BNIP3_full (plasmid) | This paper | | Mammalian expression construct encoding full-length dmBNIP3. |
| Recombinant DNA reagent | pMRX-IRES-puro-GFP-BNIP3_ΔMER (plasmid) | This paper | | Mammalian expression construct encoding ΔMER dmBNIP3. |
| Recombinant DNA reagent | LD38705 (plasmid) | DGRC | DGRC Stock Number:2722 RRID:DGRC_2722 | Atg18a cDNA |
| Recombinant DNA reagent | pMRX-IRES-puro-3xHA-mCh-Atg18a (plasmid) | This paper | | Mammalian expression construct encoding 3xHA-mCherry-dmAtg18a. |
| Sequence-based reagent | Forward primer for BNIP3 CDS | This paper | PCR primers | 5'-ATGTCTACGACA CCAAAATCGAG-3' |
| Sequence-based reagent | Reverse primer for BNIP3 CDS | This paper | PCR primers | 5'-TCAGTCAATGAC CACACGG-3' |
| Sequence-based reagent | Forward primer for plasmid construction of pUASt-attB-GFP-BNIP3_full | This paper | PCR primers | 5'-CGTGTGGTCATT GACTGAGCGGCC GCGGCTCGAGGG TACC-3' |
| Sequence-based reagent | Reverse primer for plasmid construction of pUASt-attB-GFP-BNIP3_full | This paper | PCR primers | 5'-TTGGTGTCGTAG ACATGGTGAAGG GGGCGGCCGCGG AG-3' |
| Sequence-based reagent | Forward primer for ΔLIR in BNIP3 | This paper | PCR primers | 5'-CTGCGATCGAAG CGAGCACAACAG CTGCGATG-3' |
| Sequence-based reagent | Reverse primer for ΔLIR in BNIP3 | This paper | PCR primers | 5'-CTCGCTTCGATC GCAGATTCGCCC AGCAAATC-3' |
| Sequence-based reagent | Forward primer for MER <sup>mut (L49A)</sup> in BNIP3 | This paper | PCR primers | 5'-AGACTTGCGCGC GAGGCCCAGCGC GAG-3' |
| Sequence-based reagent | Reverse primer for MER <sup>mut (L49A)</sup> in BNIP3 | This paper | PCR primers | 5'-CTCGCGCGCAAG TCTCAGGTACTCCTC-3' |
| Sequence-based reagent | Forward primer for ΔMER in BNIP3 | This paper | PCR primers | 5'-CAACAATCGCGA GTCGAACCAGTCG-3' |
| Sequence-based reagent | Reverse primer for ΔMER in BNIP3 | This paper | PCR primers | 5'-GACTCGCGATTG TTGAATGGCAACGG-3' |
| Sequence-based reagent | Upstream gRNA for BNIP3 KO | This paper | gRNA | 5'-CGTCAACACCAA AGATAACT[TGG]-3' |
| Sequence-based reagent | Downstream gRNA for BNIP3 KO | This paper | gRNA | 5'-CTTTCAGTCAAT GACCACAC[GGG]-3' |
| Sequence-based reagent | Upstream gRNA for Atg101 KO | This paper | gRNA | 5'-GTCCACCTGACG ACCCTCCA[TGG]-3' |
| Sequence-based reagent | Downstream gRNA for Atg101 KO | This paper | gRNA | 5'-CTCGCAATGTGA CGGGCTGT[CGG]-3' |

*Appendix 1 Continued on next page*

*Appendix 1 Continued*

| Reagent type (species) or resource | Designation | Source or reference | Identifiers | Additional information |
|---|---|---|---|---|
| Commercial assay or kit | ReliaPrep RNA Miniprep Systems | Promega | Cat#: Z6111 | |
| Commercial assay or kit | MEGAscript T7 kit | Thermo Fisher Scientific | Cat#: AM1334 | |
| Commercial assay or kit | RNA 6000 Pico Kit | Agilent | Cat#: 5067–1513 | |
| Commercial assay or kit | FluorSave reagent | Merck Millipore | Cat#: 345789 | |
| Commercial assay or kit | KOD One PCR Master Mix | TOYOBO | Cat#: KMM-101 | |
| Commercial assay or kit | PrimeSTAR GXL Premix | Takara Bio | Cat#: R051A | |
| Commercial assay or kit | Can Get Signal Solution 1 | TOYOBO | Cat#: NKB-201 | |
| Commercial assay or kit | Clarity Western ECL Substrate | Bio-Rad | Cat#: 1705060 | |
| Commercial assay or kit | jetPRIME transfection reagent | Polyplus-transfection | Cat#: 101000015 | |
| Chemical compound, drug | Epon 812 | TABB | Cat#: 342 | |
| Chemical compound, drug | Protease Inhibitor Cocktail | Nacalai Tesque | Cat#: 03969-34 | |
| Software, algorithm | HISAT2 | Johns Hopkins University | RRID:SCR_015530 | |
| Software, algorithm | featureCounts | PMID:24227677 | RRID:SCR_012919 | |
| Software, algorithm | UMI-tools | PMID:28100584 | RRID:SCR_017048 | |
| Software, algorithm | DESeq2 | Bioconductor | RRID:SCR_015687 | |
| Software, algorithm | pheatmap | CRAN | RRID:SCR_016418 | |
| Software, algorithm | prcomp | R base package | | |
| Software, algorithm | Mfuzz | Bioconductor | RRID:SCR_000523 | |
| Software, algorithm | FLUOVIEW | EVIDENT | | |
| Software, algorithm | ImageJ | NIH | RRID:SCR_003070 | |
| Software, algorithm | PyMOL | Schrödinger | RRID:SCR_000305 | |
| Software, algorithm | R programming language | CRAN | RRID:SCR_001905 | |
| Software, algorithm | AlphaFold Server | DeepMind | | |
| Software, algorithm | GraphPad Prism | GraphPad | RRID:SCR_002798 | Version 9.5.1 |
| Other | Alexa Fluor 633 Phalloidin | Thermo Fisher Scientific | Cat#: A22284 | (1:200). |
| Other | Bisbenzimide H33342 Fluorochrome Trihydrochloride DMSO Solution | Nacalai Tesque | Cat#: 04915-81 | (1:10,000 dilution) |
| Other | GFP-Trap Agarose | Chromotek | Cat#: gta | |

