## [Editor Report · eLife Assessment]

This paper presents the **important** finding that BNIP3/NIX, a mitophagy receptor, and its binding to ATG18 are required for mitophagy during muscle cell reorganization in Drosophila. Although the involvement of the BNIP3-ATG18/WIPI axis in mitophagy induction has been reported in mammalian cell culture systems, this study provides the first **compelling** evidence for this pathway in vivo in animals. The physiological significance of this BNIP3-dependent mitophagy will require further investigation.

---

## [Referee Report · Reviewer #1 (Public review)]

During early Drosophila pupal development, a subset of larval abdominal muscles (DIOMs) is remodelled using an autophagy dependent mechanism.

To better understand this not very well studied process, the authors have generated a systematic transcriptomics time course using dissected larval abdominal muscles of various stages from wild type and autophagy deficient mutants. The authors have further identified a function for BNIP3 for executing mitophagy during DIOM remodelling.

Strengths:

The paper does provide a detailed mRNA time course resource for the DIOM remodelling.

The paper does find an interesting BNIP3 loss of function phenotype, a block of mitophagy during muscle remodelling and hence identifies a specific linker between mitochondria and the core autophagy

machinery. This adds to the mechanism how mitochondria are degraded.

Sophisticated fly genetics demonstrates that the larval muscle mitochondria are, to a large extend, degraded by autophagy during DIOM remodelling.

Quantitative electron microscopy data show that BNIP3 is required for initiating mito-phagosomes. It needs either its LIR and MER domain for function.

Weakness:

Mitophagy during DIOM remodelling is not novel (earlier papers from Fujita et al.).

Other weaknesses have been eliminated during the revision.

---

## [Referee Report · Reviewer #2 (Public review)]

Summary:

Autophagy (macroautophagy) is known to be essential for muscle function in flies and mammals. To date, many mitophagy (selective mitochondrial autophagy) receptors have been identified in mammals and other species. While loss of mitophagy receptors has been shown to impair mitochondrial degradation (e.g., OPTN and NDP52 in Parkin-mediated mitophagy and NIX and BNIP3 in hypoxia-induced mitophagy) at the level of cultured cells, it remains unclear, especially under physiological conditions in vivo. In this study, the authors revealed that one of the receptors BNIP3 plays a critical role in mitochondrial degradation during muscle remodeling in vivo.

Overall, the manuscript provides solid evidence that BNIP3 is involved in mitophagy during muscle remodeling with in vivo analyses performed. In particular, all experiments in this study are well designed. The text is well written and the figures are very clear.

Strengths:

(1) In each experiment, appropriate positive and negative controls are used to indicate what is responsible for the phenomenon observed by the authors: e.g. FIP200, Atg18, Stx17 siRNAs during DIOM remodeling in Fig2 and Full, del-LIR, del-MER in Fig5.

(2) Although the transcriptional dynamics of DIOM remodeling during metamorphosis is autophagy-independent, the transcriptome data obtained by the authors would be valuable for future studies.

(3) In addition to the simple observation that loss of BNIP3 causes mitochondrial accumulation, the authors further observed that, by combining siRNA against STX17, which is required for fusion of autophagosomes with lysosomes, BNIP3 KO abolishes mitophagosome formation, which will provide solid evidence for BNIP3-mediated mitophagy. Furthermore, using a Gal80 temperature-sensitive approach, the authors showed that mitochondria derived from larval muscle, but not those synthesized during hypertrophy, remain in BNIP3 KO fly muscles.

Weaknesses:

(1) Because BNIP3 KO causes mitochondrial accumulation, it is expected that adult flies will have some physiological defects, but this has not been fully analyzed or sufficiently mentioned in the manuscript.

(2) In Fig 5, the authors showed that BNIP3 binds to Atg18a by co-IP, but no data are provided on whether MER-mut or del-MER attenuates the affinity for Atg18a.

Comments on revisions: The authors answered all the reviewer's concerns.

---

## [Referee Report · Reviewer #3 (Public review)]

Summary:

Fujita et al build on their earlier, 2017 eLife paper that showed the role of autophagy in the developmental remodeling of a group of muscles (DIOM) in the abdomen of Drosophila. Most larval muscles undergo histolysis during metamorphosis, while DIOMs are programmed to regrow after initial atrophy to give rise to temporary adult muscles, which survive for only 1 day after eclosion of the adult flies (J Neurosci. 1990;10:403-1. and BMC Dev Biol 16, 12, 2016). The authors carry out transcriptomics profiling of these muscles during metamorphosis, which are in agreement with the atrophy and regrowth phases of these muscles. Expression of the known mitophagy receptor BNIP3/NIX is high during atrophy, so the authors start to delve more into the role of this protein/mitophagy in their model. BNIP3 KO indeed impairs mitophagy and muscle atrophy, which they convincingly demonstrate via nice microscopy images. They also show that the already known Atg8a-binding LIR and Atg18a-binding MER motifs of human NIX are conserved in the Drosophila protein, although the LIR turned out to be less critical for in vivo protein function than the MER motif.

Strengths:

Established methodology, convincing data, in vivo model

Weaknesses:

Significance for Drosophila physiology and for human muscles remains to be established

---

## [Author Response]

The following is the authors’ response to the original reviews.

**Reviewer #1 (Public review):**
Summary:During early *Drosophila* pupal development, a subset of larval abdominal muscles (DIOMs) is remodelled using an autophagy-dependent mechanism.To better understand this not very well studied process, the authors have generated a transcriptomics time course using dissected abdominal muscles of various stages from wild-type and autophagy-deficient mutants. The authors have further identified a function for BNIP3 in muscle mitophagy using this system.Strengths:(1) The paper does provide a detailed mRNA time course resource for DIOM remodeling.(2) The paper does find an interesting BNIP3 loss of function phenotype, a block of mitophagy during muscle remodeling, and hence identifies a specific linker between mitochondria and the core autophagy machinery. This adds to the mechanism of how mitochondria are degraded.(3) Sophisticated fly genetics demonstrates that the larval muscle mitochondria are, to a large extent, degraded by autophagy during DIOM remodeling.Weaknesses:(1) Mitophagy during DIOM remodeling is not novel (earlier papers from Fujita et al.).(2) The transcriptomics time course data are not well connected to the autophagy part. Both could be separated into 2 independent manuscripts.(3) The muscle phenotypes need better quantifications, both for the EM and light microscopy data in various figures.(4) The transcriptomics data are hard to browse in the provided PDF format.

Thank you for reviewing our manuscript and for your feedback. While we understand and appreciate the suggestion to divide the manuscript into two separate studies, we believe that presenting the work as a single manuscript is more appropriate. This is because the time-course RNA-seq of DIOMs provides critical insight into BNIP3-mediated mitophagy during DIOM remodeling, which ties together the two components of our study. In response to Reviewer #1’s recommendations, we have quantified data from both EM and confocal images, and we have revised the RNA counts table in Supplementary File 1 accordingly. Please see our detailed responses and revisions on the following pages.

**Reviewer #2 (Public review):**
Summary:Autophagy (macroautophagy) is known to be essential for muscle function in flies and mammals. To date, many mitophagy (selective mitochondrial autophagy) receptors have been identified in mammals and other species. While the loss of mitophagy receptors has been shown to impair mitochondrial degradation (e.g., OPTN and NDP52 in Parkin-mediated mitophagy and NIX and BNIP3 in hypoxia-induced mitophagy) at the level of cultured cells, it remains unclear, especially under physiological conditions in vivo. In this study, the authors revealed that one of the receptors BNIP3 plays a critical role in mitochondrial degradation during muscle remodeling in vivo.Overall, the manuscript provides solid evidence that BNIP3 is involved in mitophagy during muscle remodeling with in vivo analyses performed. In particular, all experiments in this study are well-designed. The text is well written and the figures are very clear.Strengths:(1) In each experiment, appropriate positive and negative controls are used to indicate what is responsible for the phenomenon observed by the authors: e.g. FIP200, Atg18, Stx17 siRNAs during DIOM remodeling in Figure 2 and Full, del-LIR, del-MER in Figure 5.(2) Although the transcriptional dynamics of DIOM remodeling during metamorphosis is autophagy-independent, the transcriptome data obtained by the authors would be valuable for future studies.(3) In addition to the simple observation that loss of BNIP3 causes mitochondrial accumulation, the authors further observed that, by combining siRNA against STX17, which is required for fusion of autophagosomes with lysosomes, BNIP3 KO abolishes mitophagosome formation, which will provide solid evidence for BNIP3-mediated mitophagy. Furthermore, using a Gal80 temperature-sensitive approach, the authors showed that mitochondria derived from larval muscle, but not those synthesized during hypertrophy, remain in BNIP3 KO fly muscles.Weaknesses:(1) Because BNIP3 KO causes mitochondrial accumulation, it is expected that adult flies will have some physiological defects, but this has not been fully analyzed or sufficiently mentioned in the manuscript.(2) In Figure 5, the authors showed that BNIP3 binds to Atg18a by co-IP, but no data are provided on whether MER-mut or del-MER attenuates the affinity for Atg18a.

Thank you for pointing out the critical issues in the previous version of our manuscript. In this revision, we have conducted several physiological assays using BNIP3 KO flies, as well as co-IP experiments to confirm that the DMER weakens the interaction with Atg18a. We have also addressed all the recommendations provided. Please see our detailed point-by-point responses below.

**Reviewer #3 (Public review**):Summary:Fujita et al build on their earlier, 2017 eLife paper that showed the role of autophagy in the developmental remodeling of a group of muscles (DIOM) in the abdomen of *Drosophila*. Most larval muscles undergo histolysis during metamorphosis, while DIOMs are programmed to regrow after initial atrophy to give rise to temporary adult muscles, which survive for only 1 day after eclosion of the adult flies (J Neurosci. 1990;10:403-1. and BMC Dev Biol 16, 12, 2016). The authors carry out transcriptomics profiling of these muscles during metamorphosis, which is in agreement with the atrophy and regrowth phases of these muscles. Expression of the known mitophagy receptor BNIP3/NIX is high during atrophy, so the authors have started to delve more into the role of this protein/mitophagy in their model. BNIP3 KO indeed impairs mitophagy and muscle atrophy, which they convincingly demonstrate via nice microscopy images. They also show that the already known Atg8a-binding LIR and Atg18a-binding MER motifs of human NIX are conserved in the Drosophila protein, although the LIR turned out to be less critical for in vivo protein function than the MER motif.Strengths:Established methodology, convincing data, in vivo model.Weaknesses:The significance for *Drosophila* physiology and for human muscles remains to be established.

Thank you for reviewing our manuscript. In response to the comment, we have performed lifespan, adult locomotion, and eclosion assays in *BNIP3* KO flies. Although we observed substantial mitochondrial accumulation in the DIOMs of *BNIP3* KO flies, no significant differences were detected in these physiological assays under our experimental conditions. We plan to further investigate the physiological role of BNIP3 in flies and extend our studies to human muscle in future work. Please see our detailed responses below.

**Reviewer #1 (Recommendations for the authors):**
Major points:(1) Unfortunately, the RNA counts file table in Supplementary file 1 is a PDF and not an Excel sheet. The labelling makes it unclear from which time points and genotype the listed values on the 650-page files are.

We have now corrected the labelling of time points and genotypes in Supplementary File 1 to improve clarity and have provided the updated Excel file.

Looking at these counts it seems that sarcomere genes (Mhc, bt, sls, wupA, TpnC) are 10x to 100x lower in sample "ctrl_1" compared to the three other control samples. Which time point is that? It is essential to have access to the full dataset, wild type and autophagy-deficient, to be able to assess the quality of the RNA SEQ data. These need to be deposited in a public database or to be provided in a useful format.

Thank you for pointing that out. In the previous version, “Ctrl_1” referred to the Control sample at 1 day APF, when atrophy occurs. We have corrected the labeling in Supplementary File 1 accordingly and have deposited the RNA-seq data to GEO, where it is now publicly available (GSE293359).

(2) Which statistical test was used to assess the differences in muscle volumes in Figure 2E? I was not able to find a table with the measured data.

In Figure 2E, we used the Mann-Whitney test for statistical analysis. The raw data used for quantification have also been provided (Supplementary File 2).

The shown volumes do not correlate with the scheme shown in Figure 2A, in particular at the larval stage the muscle seems much larger.

We have revised the schematic models of muscle cells in Figures 1C and 2A in accordance with the reviewer’s suggestion.

(3) It is important to remember that adult *Drosophila* muscles are not homogenous, at least not the adult leg and abdominal muscles, as they are organised as tubes with myofibrils closer to the surface, and nuclei as well as mitochondria largely in the centre (see PMID 33828099). Hence, only showing a single plane in the muscle images can be very misleading. The authors should at least provide virtual XZ-cross section views in Figure 3G to ensure that similar muscle planes are compared. This applies to the interpretation of both, the mitochondria and the myofibril phenotypes in wildtype vs BNIP3-KO.

Thank you for your comment. As suggested, we have added XZ-cross-sectional views in Figure 3G. The XY plane corresponds to a central section of the Z-stack, as indicated in the figure.

(4) The EM images are nice, however only 2 of the 4 conditions shown were quantified. As the section plane can be misleading, at least several planes should be analysed also for wild type and BNIP3-KO, and not only for stx17 RNAi and the double mutant.

In response to the comment, we quantified the TEM images of wild-type and BNIP3-KO DIOMs and added the resulting graph to Figure 4C. The corresponding raw data have also been provided (Supplementary File 2).

(5) How was Figure 5D, 5D' quantified? What corresponds to "regular", "medium", "high"? A statistical test is missing. I would rather conclude that MIR and LIR are redundant as double mutant appears to be stronger than both singles. This is also concluded in some sections of the text, so the authors seem to contradict themselves. Why not measure the mitochondria areas as done in Figure 6A' instead?

In the previous version, we manually categorized pooled, blinded images from different genotypes. However, as the reviewer pointed out, this approach was not quantitative. In the revised version, we analyzed the images using ImageJ to quantify the mitochondrial area per cell. Statistical significance was assessed using the Kruskal-Wallis test. Accordingly, we have revised Figure 5D, the method section, and the figure legend.

(6) Figure 6B data seem to come from a single image per genotype only. At least 3 or 4 animals should be measured and the values reported.

We analyzed Pearson’s correlation coefficients (R values) from at least five images per genotype and performed statistical analysis. The resulting quantification is presented in Figure 6B’, and the corresponding text has been revised accordingly.

(7) As BNIP3 mutants are viable, it would be interesting to report if they can fly and how long they live.

Additional data on adult lifespan, climbing ability, and elapsed time for eclosion in *BNIP3* KO flies have been included as supplemental information (Figure 3-figure supplement 2). No significant differences were observed in those assays under our experimental conditions.

(8) The transcriptomics data are not well linked to the autophagy mechanism. In particular, the mutant transcriptomics data are confusing, as the abstract seems to suggest that blocking autophagy impacts transcriptomics, which is not (strongly) the case. I would at least re-write this part, as it is currently misleading and sparks wrong expectations to the reader. Also throughout the text, the authors need to make clear if there are transcriptomic changes or not and if there are, how these are linked to autophagy.

In the abstract, we described the findings as “transcriptional dynamics independent of autophagy” (line 49) because the loss of autophagy had only a minimal effect on transcriptional changes. This conclusion is supported by the data presented in our manuscript. In the result section, we state: “In contrast to our prediction, the knockdown of *Atg18a*, *FIP200*, or *Stx17* only had a slight impact on transcriptomic dynamics in DIOM remodeling (Fig. 2C), with only minor changes detected (Fig. 2-figure supplement 2G)” (lines 199-201). In the Discussion section, we further note: “The transcriptional dynamics associated with DIOM remodeling are largely independent of autophagy (Fig.2). Instead, our RNA-seq data suggest that it is regulated primarily by ecdysone signaling, with minimal influence from autophagy inhibition” (lines 326-328).

(9) No table with the measured data is provided.

We have provided the raw data files corresponding to all quantified results as Supplementary File 2.

Minor points:
*(1) To my knowledge, it is standard to indicate the time after puparium formation in hours, instead of days, (e.g. 24h, 48h etc.).*

Thank you for the comments. In our previous publications on DIOM remodeling during metamorphosis (PMID: 28063257 and 33077556), we used days rather than hours to indicate developmental time points. To maintain consistency across our studies, we have chosen to continue using days in the present manuscript.

(2) "Myofibrils typically form beneath the sarcolemma (Mao et al., 2022; Sanger et al., 2010); therefore, when mitochondria accumulate, myofibrils are restricted to the cell periphery." This is quite a general statement that does not always hold, in particular not in *Drosophila* flight muscles and likely also not in abdominal muscles (see PMIDs 29846170, 28174246).

Thank you for pointing that out. We rewrote the sentence as follows: In the absence of BNIP3, mitochondria derived from the larval muscle accumulate and cluster in the cell center, physically obstructing myofibril formation during hypertrophy and restricting myofibrils to the cell periphery (Fig. 6E) (lines 392-394).

**Reviewer #2 (Recommendations for the authors):**
Suggestions for improved or additional experiments, data or analyses.The authors should test, by a co-IP experiment, whether BNIP3 mutants lose the interaction with HA-Atg18a.

As requested, we tested the effect of MER deletion on the interaction between BNIP3 and Atg18a in co-IP experiment. As shown in the new Fig. 5C, the deletion of MER weakened the interaction. This result was confirmed in three independent experiments. Its corresponding text has also been revised as follows: “We confirmed that HA-tagged *Drosophila* Atg18a co-immunoprecipitated with GFP-tagged full-length *Drosophila BNIP3*, and that this interaction was attenuated by the deletion of the MER (residues 42-53) (Fig. 5C)” (lines 270-273).

Minor corrections to the text and figures(1) In the list of authors, Kawaguchi Kohei could be Kohei Kawaguchi_._

Thank you very much. It has been corrected.

(2) In Fig3D, other receptors (Zonda, CG12511, Key, Ref2P) should be mentioned briefly.

Thank you for the suggestion. We have revised the sentences as follows: “The time course RNA-seq data (Fig. 1 and 2) indicated that, among the known mitophagy regulators, only BNIP3 was robustly expressed in 1 d APF DIOMs. In contrast, *Zonda*, *CG12511*, *Pink1*, *Park*, *Key*, *Ref(2)P*, and *IKKe*—the *Drosophila* orthologs of FKBP8, FUNDC1, PINK1, Parkin, Optineurin, p62, and TBK1, respectively—showed little or undetectable expression at this stage (Fig. 3D).” (lines 230-234).

**Reviewer #3 (Recommendations for the authors):**
Remarks:(1) What is the consequence of impaired muscle remodeling on the organismal level? Is the eclosion of adult flies impaired? One could think of assays for this, such as quantifying failed eclosions and/or video microscopy of the eclosion process. Is muscle function impaired? One could measure the contractile force of isolated fibers during electrical stimulation as well, etc. I believe that showing the physiological importance of muscle remodeling would be the biggest advantage that could arise from using a complete animal model.

We appreciate the comments. We have added data on adult lifespan, climbing ability, and the elapsed time for eclosion in *BNIP3* KO flies as supplemental information (Figures 3-figure supplement 2). In *BNIP3* KO DIOMs, despite the massive accumulation of mitochondria, an organized peripheral myofibril layer with contractile function is retained. However, we have not measured the contractile force of isolated muscle cells due to technical limitations. We plan to address this in future studies.

A related note is that I missed the proper discussion of the function and fate of these short-lived adult muscles (please see references in my summary).

We have added a sentence regarding the function and fate of DIOMs in the introduction (lines 80-82) as follows: “The remodeled adult DIOMs function during eclosion, persist for approximately 12 hours, and are subsequently eliminated via programmed cell death (Kimura and Truman, 1990; J Neurosci. 1990;10:403-1)”.

(2) I don't think that "data not shown" should be used these days, when supplemental data allow the inclusion of not-so-critical results.

We have added the data as Figure 5-figure supplement 2. As shown in the figure, overexpression of GFP-BNIP3 in 3IL BWMs did not induce the formation of tdTomato-positive autolysosomes, which are abundantly accumulated in DIOMs at 1 and 2 d APF.

(3) The term "naked mitochondria" does not sound scientific enough to this reviewer. I suggest "cytosolic mitochondria" or "unengulfed mitochondria".

In accordance with the reviewer’s suggestion, we have replaced “naked mitochondria” with “unengulfed mitochondria” (lines 251 and 670).